# Adversarial Attacks and Defenses in Vision-Language Pre-training: Techniques, Challenges and Opportunities

## Abstract

Vision-language pretraining (VLP) has emerged as a powerful paradigm for multimodal learning. However, despite their superior capabilities, VLPs remain vulnerable to adversarial attacks by manipulating their inputs. Such attacks by undermining user trust can significantly compromise their integrity, introduce critical security vulnerabilities and highlight the importance of securing VLPs to ensure safety in various real-world multimodal applications. In the adversarial landscape of VLPs, this review aims to delve into the methodologies and implications of both adversarial attacks and defense strategies, organized by architectural considerations. Our review delves into the complexities of categorizing adversarial attack strategies, underscoring the critical need for robust defensive measures. To improve the reliability of these models, we discuss novel defense mechanisms that counter vulnerabilities. In addition, we analyze how adversarial vulnerabilities impact downstream applications. Overall, this review aims to provide a comprehensive overview of adversarial threats in VLPs and present future research directions.

## 1 Introduction

### 1.1 VLP Adversarial Attack Scenario

Vision-language pretraining models (VLPs), inspired by the success of transformers (Ngiam et al., 2011; Long et al., 2022), have revolutionized the landscape of multimodal machine learning tasks. These VLPs (Du et al., 2022), which integrate Computer Vision (CV) and Natural Language Processing (NLP), are typically pre-trained on large-scale datasets consisting of both visual and textual modalities. This paradigm enables strong multimodal understanding through rich cross-modal representations, thereby supporting a wide range of applications such as visual question answering (VQA), cross-modal retrieval, dialogue systems, image captioning, and content explanation. VLPs are increasingly deployed in security-sensitive applications, including healthcare (Finlayson et al., 2019), autonomous driving (Cao et al., 2019), and interdependent AI systems (Zhao et al., 2024b), marking a key shift in the current AI technological revolution. However, VLPs remain vulnerable to adversarial attacks due to the dependence on large-scale Internet datasets for pre-training. These risks extend beyond academic interest into real-world scenarios, as VLPs are increasingly deployed in practical applications. In high-stakes scenarios, adversarial failures can lead to serious safety risks, misclassifications, or biased decision-making, thereby undermining user trust and limiting the broader adoption of VLP systems.

Adversarial attacks on VLPs (Nakano et al., 2024) have become increasingly sophisticated and diverse, targeting single or multiple modalities to induce multimodal misinterpretations and lead to erroneous predictions in complex AI systems. These findings have motivated extensive research into exploring the reliability and safety of VLPs under adversarial settings, prompting the development of attack and defense technologies. Compared to unimodal models, VLPs face unique security challenges due to their multimodal nature, as traditional adversarial attacks in CV focus on perturbing images at the pixel level and in NLP concentrate on token-level or embedding-level perturbations. Given these complexities, research into the adversarial robustness of VLPs has aimed to systematically examine the multimodal vulnerabilities arising from the joint modeling of vision and language. Understanding adversarial robustness in VLPs is essential for uncovering previously unexplored vulnerabilities and addressing emerging interdisciplinary threats. Ultimately, developing robust VLP systems requires a comprehensive examination of such security properties to enhance their reliability and safety in security-sensitive applications.

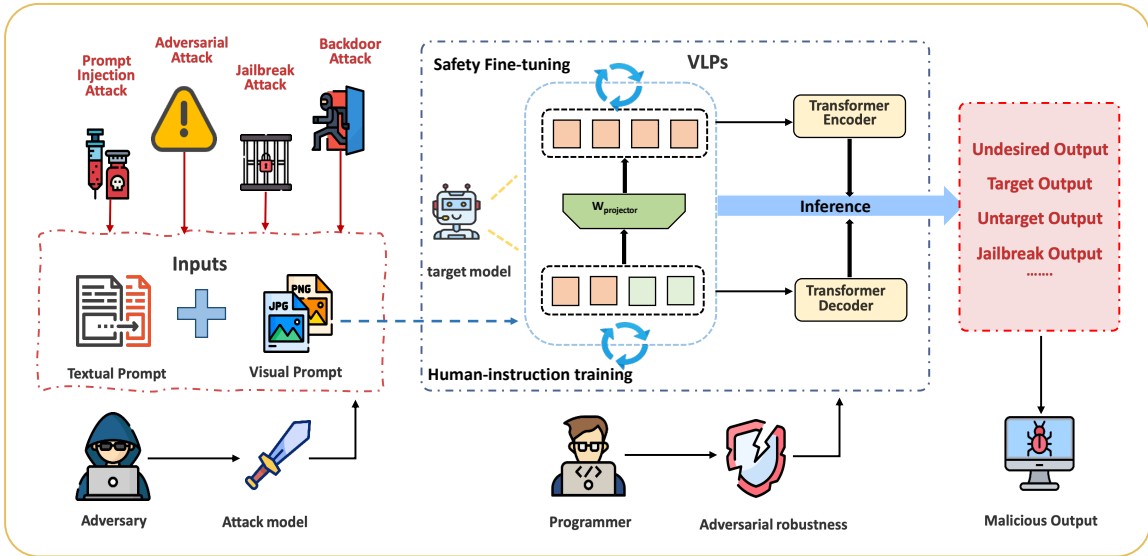

Figure 1: The figure illustrates how adversaries can manipulate textual and visual inputs through prompt injection attacks, adversarial attacks, jailbreak attacks and backdoor attacks to VLPs

Adversarial attacks for VLPs deliberately target inputs across different modalities to mislead models, posing a significant threat to VLP robustness. Given this threat, adversarial attacks and defenses for VLPs have received increasing research attention (Nakano et al., 2024). However, despite this growing interest, there is a lack of a systematic review that comprehensively examines these attack and defense strategies for VLPs. To the best of our knowledge, this review provides the comprehensive analysis of adversarial exploitation in VLPs. In this review, we categorize these threats and defenses as shown in Figure 1. The attacks include prompt injection, adversarial perturbations, jailbreak attacks, and backdoor attacks. Correspondingly, defense strategies include adversarial contrastive fine-tuning, adversarial prompt tuning, backdoor defense, jailbreak defense, and safety alignment.

## 1.2 Contributions and Motivation for This Review

VLPs operate at the convergence of visual and linguistic processing, offering a promising paradigm for multimodal understanding and generation. They increasingly encounter adversarial threats that compromise their reliability and security across various multimodal applications. Such vulnerabilities motivate us to undertake a comprehensive investigation into adversarial robustness in VLPs. To bridge this gap, we propose a taxonomy that systematically categorizes adversarial attack and defense techniques based on VLP architectural perspectives. Through this work, we strive to raise awareness of AI security within the research community, while providing a stepping-stone for advancing trustworthy vision-language learning. Our main contributions are as follows:

- We propose a structured taxonomy that categorizes current VLPs based on their architectures and training objectives.

- We comprehensively summarize recent adversarial attack and defense techniques for VLPs, analyzing their effectiveness, advantages, and limitations across different model types.

- We provide an in-depth discussion of key challenges and open problems in VLP adversarial robustness, highlighting promising future research directions.

| Survey | Focus Area | Core Topic | Attack Types Covered | Defense Types Covered | Vision-Language Specific |
|---|---|---|---|---|---|
| Shayegani et al. (2023a) | LLMs | Attack & Defense | Adversarial Attacks, Jailbreak Attacks | Detection, Prevention, Safety Alignment | Partial |
| Vatsa et al. (2024) | VLPs | Bias, Robustness, Interpretability | Adversarial Attacks | Robustness Enhancement | ✓ |
| Jin et al. (2024) | LLMs/LVLMs | Jailbreaking | Jailbreak Attacks | Jailbreak Defenses | Partial |
| Liu et al. (2024b) | LVLMs | Attacks | Adversarial Attacks, Jailbreak Attacks, Prompt Injection Attacks, Backdoor Attacks | ✗ | ✓ |
| Dang et al. (2024) | LVLMs | Interpretability | ✗ | ✗ | ✓ |
| Zhang et al. (2024g) | LVLMs | Trustworthiness Evaluation | Adversarial Attacks, Jailbreak Attacks | ✗ | ✓ |
| **Ours** | **VLPs** | **Attack & Defense** | **Adversarial Attacks, Jailbreak Attacks, Prompt Injection and Backdoor Attacks** | **Adversarial Defenses, Jailbreak Defenses, Backdoor Defenses, Certified Robustness** | ✓ |

Table 1: Comparison between our survey and related surveys on the trustworthiness and security of VLPs. "Partial" indicates that vision-language models are included but are not the exclusive focus, while ✗ indicates that the corresponding topic is not systematically covered.

We propose an architecture-aware taxonomy for adversarial robustness in VLPs. This taxonomy organizes adversarial attacks and defenses according to three representative VLP paradigms: vision-language fusion pretraining (VLFP), vision-language contrastive learning (VLCL), and large vision-language models (LVLMs). This catalogue highlights their architectures and training objectives, connecting each architecture with representative attack mechanisms and corresponding defense challenges. In contrast to AutoDAN (Liu et al., 2024f) focusing on LLM jailbreak prompts classification and Adversarial Illusions (Zhang et al., 2024f) studying to attack multimodal embedding spaces. our taxonomy is organized around VLP architecture families and the adversarial scenarios specific to each.

## 1.3 Comparison to Existing Surveys on Adversarial Robustness in VLPs

While prior works have surveyed adversarial attacks and defenses for Large Language Models (LLMs) and Large Vision-Language Models (LVLMs), Limited work has provided a comprehensive analysis specifically focused on VLP types and their adversarial robustness. Table 1 compares our review with existing literature regarding scope, focus and coverage. We summarize the key differences as follows:

- **Shayegani et al. (2023a):** Summarize adversarial attacks and defenses on Large language Model (LLM) and Large Vision-language Model (LVLM). They provide a taxonomy of existing attacks based on single modality, multiple modalities and additional attack types with threat model scenarios.

- **Vatsa et al. (2024):** Review the trustworthiness of VLPs from three aspects: bias, robustness, and interpretability, with a particular focus on VLP architectures and downstream tasks.

- **Jin et al. (2024):** Provide a detailed explaination of jailbreaking attacks for LLMs, categorizing seven jailbreak types and corresponding defense strategies, with brief coverage of LVLMs.

- **Liu et al. (2024b):** Discuss security vulnerabilities in LVLMs, covering adversarial attacks, jailbreak attacks, prompt injection attacks, and backdoor attacks.

- **Dang et al. (2024):** Focus on interpretability and explainability of LVLMs for trustworthiness, analyzing existing research from data, model, and training/inference perspectives.

- **Zhang et al. (2024g):** Present a unified benchmark for evaluating LVLM trustworthiness across five aspects: truthfulness, safety, robustness, fairness and privacy, covering various attack scenarios and bias issues.

- **Liu et al. (2023b):** Provide a survey of trustworthy machine learning that reviews robustness, security, interpretability and fairness under a data-centric perspective, and extends this perspective to causal inference methods and frameworks for large pretrained models.

Based on the aforementioned works, this review distinguishes itself from existing studies in several key aspects. First, we provide a comprehensive analysis of various VLP architectures, along with their training

| Abbreviation | Full Form | Abbreviation | Full Form |
|---|---|---|---|
| 3VL | Tree-Augmented Vision-Language | ACL | Adversarial Contrastive Learning |
| AI | Artificial Intelligence | APGD | Auto-PGD |
| ALIP | Adaptive Language–Image Pre-training | BaThe | Backdoor Trigger Shield |
| APT | Adversarial Prompt Tuning | CLAP | Contrastive Learning with Augmented Prompts |
| C-AVP | Class-wise Adversarial Visual Prompting | CLIP | Contrastive Language-Image Pre-Training |
| CNN | Convolutional Neural Network | CV | Computer Vision |
| ECSO | Eyes Closed and Safety On | EOS | End-of-Sequence |
| FAP | Few-shot Adversarial Prompt Learning | FARE | Fine-tuning for Adversarially Robust Embeddings |
| GAN | Generative Adversarial Networks | GPT | Generative Pre-trained Transformer |
| IC | Image Captioning | ITR | Image Text Retrieval |
| LLM | Large Language Model | LVLM | Large Vision-Language Model |
| MMT | Multimodal Machine Translation | MTG | Multimodal Text Generation |
| NLG | Natural Language Generation | NLP | Natural Language Processing |
| NLU | Natural Language Understanding | NLVR | Natural Language for Visual Reasoning |
| NoCaps | Novel Object Caption at Scale | OCR | Optical Character Recognition |
| ORCA | Observe–Reason–Critique–Act | OVC | Open Vocabulary Certification |
| PGD | Projected Gradient Descent | PSA-VLM | Progressive Safety Alignment for Vision-Language Models |
| RAN | Rectify Adversarial Noise | SVM | Support Vector Machine |
| TAPT | Test-Time Adversarial Prompt Tuning | TIJO | Trigger Inversion using Joint Optimization |
| UMK | Universal Master Key | VD | Visual Dialogue |
| VE | Visual Entailment | VG | Visual Grounding |
| ViT | Vision Transformer | VCR | Visual Commonsense Reasoning |
| VLCL | Vision-Language Contrastive Learning | VLFP | Vision-Language Fusion Pretraining |
| VLN | Visual Linguistic Navigation | VLP | Vision-Language Pretraining |
| VQA | Visual Question Answering | VR | Visual Reasoning |

Table 2: Abbreviations and Definitions

process and targets, filling the gap in existing literature that primarily focuses on LLMs or LVLMs. Second, we systematically investigate cross-modal vulnerabilities and the unique security challenges posed by multimodal interactions, analyzing corresponding attack and defense strategies through our proposed VLP taxonomy. Third, we analyze attack and defense challenges within each category of our taxonomy. This analysis identifies key trends, highlights existing limitations, and uncovers potential solutions that have been investigated in the previous works.

We conduct a systematic literature review spanning multiple research domains, including NLP, CV, machine learning, and multimodal learning. We focus on publications from January 2020 to September 2025 to capture the most recent advancements in adversarial robustness for vision-language models. Our search strategy encompasses premier venues, including top-tier conferences such as NeurIPS, ICML, ICLR, CVPR, ICCV, ECCV, ACL, EMNLP, AAAI, IJCAI, and leading journals such as IEEE TPAMI, IEEE TIP, JMLR, TMLR and ACM Computing Surveys. We further supplement this process with comprehensive searches using Google Scholar, arXiv, and major digital libraries to ensure broad coverage of relevant literature. We adopt a three-stage screening methodology: (1) title-based initial filtering to identify potentially relevant studies; (2) abstract-based assessment to evaluate direct relevance to adversarial robustness in VLP; and (3) full-text review for final inclusion decisions when abstracts provide insufficient information.

Our search queries combined vision-language terms (e.g.,"vision-language pre-training," "CLIP," and "large vision-language models") with security-related terms (e.g., "adversarial attack," "jailbreak," "backdoor," "data poisoning," and "adversarial defense"). We included studies adversarial attacks, adversarial defenses, robustness evaluation, or certification for VLPs; consider visual, textual, or cross-modal mechanisms; and provide sufficient methodological or experimental details for analysis. This review contains 457 reference and the core synthesis includes 67 studies on VLP architectures, 116 attack-related studies, and 107 defense or evaluation studies of VLPs. This systematic approach ensures comprehensive coverage while maintaining high relevance standards. The remaining references provide background on Transformer architectures, visual and language representation learning, datasets, downstream tasks, and general multimodal learning. Table 2 provides definitions for all abbreviations used throughout this review.

As illustrated in Fig. 2, the remainder of this survey is organized as follows: Section 2 reviews recent advances in VLPs. Section 3 introduces vision-language downstream tasks and examines the impact of adversarial attacks on their performance. Section 4 introduces the taxonomy of VLP architectures and adversarial threats. Section 5 categorizes adversarial attacks against VLPs and analyzes each category. Section 6 reviews adversarial defense strategies and discusses their effectiveness and limitations. Finally, Section 7 discusses the main challenges and future research directions for improving the adversarial robustness of VLPs.

## 2 Vision-Language Pretraining Paradigms

Modern VLP frameworks have demonstrated remarkable effectiveness in integrating text and image pretraining for multimodal tasks (Mogadala et al., 2021; Long et al., 2022). This VLP paradigm excels at extracting features from complex high-dimensional data by leveraging the ability to model long-range dependencies and integrate information across modalities. VLP training strategies typically combine supervised learning with large-scale self-supervised pretraining, enhancing cross-modal representations through specialized pretraining objectives and advanced architectures. This approach enables models to learn semantic correspondences across modalities, benefiting downstream tasks while avoiding training from scratch. The current tendency in this field has been moving toward developing larger models and utilizing more extensive pretraining datasets (Chen et al., 2023b). However, existing VLPs exhibit diverse architectural frameworks and lack built-in security protections, exposing security risks under adversarial conditions. To systematically analyze adversarial robustness across different model types, we categorize VLP architectures from three perspectives: Vision-Language Fusion Pretraining (VLFP), Vision-Language Contrastive Learning (VLCL) and LVLMs, as illustrated in Figure 3. We also summarize representative VLP models in Table 3, highlighting their differences in architectural design and training objectives.

### 2.1 Vision-language Fusion Pretraining

VLFP represents a prevalent approach in multimodal fusion, comprising three core components: vision encoders, language encoders and multimodal fusion modules. Vision Encoder, also known as visual feature extractor, uses Convolutional Neural Network (CNN) for Grid Features (He et al., 2016), Object Detection for Region Features (Ren et al., 2015), Vision Transformer (ViT) (Dosovitskiy et al., 2021) and CLIP-ViT (Radford et al., 2021) for patch features. Language Encoder, also known as linguistic feature extractor, generates textual representations by processing input sequences with special tokens. Pre-trained language models such as BERT (Devlin et al., 2019), RoBERTa (Liu et al., 2019) and DeBERTa (He et al., 2020) are typical examples of such encoders and have demonstrated significant improvements in NLP tasks. The Multimodal Fusion Module integrates the encoded image and text features within the same semantic space using the attention mechanism to enable cross-modal interactions. These architectural designs enable the complementary strengths of visual and linguistic representations, enhancing cross-modal representations to achieve remarkable success in multimodal understanding tasks.

The VLFPs have been widely adopted in various vision-language tasks, with their fusion architectures generally categorized into two common types: single-stream and dual-stream (Cho et al., 2021). The single-stream aligns with the encoding structure of the Transformer framework, to integrate visual and textual information directly concatenated into an unified representation. For instance, VisualBERT (Li et al., 2020a) proposes a flexible framework that uses object detection as visual features for image-text fusion without explicit supervision. Oscar (Li et al., 2020b) presents an object-semantics alignment approach that uses object tags recognized in images as anchor points to assemble text-tag-image triples for enhanced fusion, though this inadvertently introduces redundancy and attribute noise. VinVL (Zhang et al., 2021a) extends this idea by developing a new object detection model that incorporates the object attribute into image encoding. This learning mode has enhanced visual representations at the cost of increased model size and computational overhead. Instead of replying on pre-trained object detectors, ViLT (Kim et al., 2021) simplifies the architecture by directly integrating image features through computationally efficient ViT encoder with BERT for linguistic features. UNITER (Chen et al., 2020) employs conditional masking strategies to mask either the image or text (one modality at a time). This is opposed to the joint random masking on both

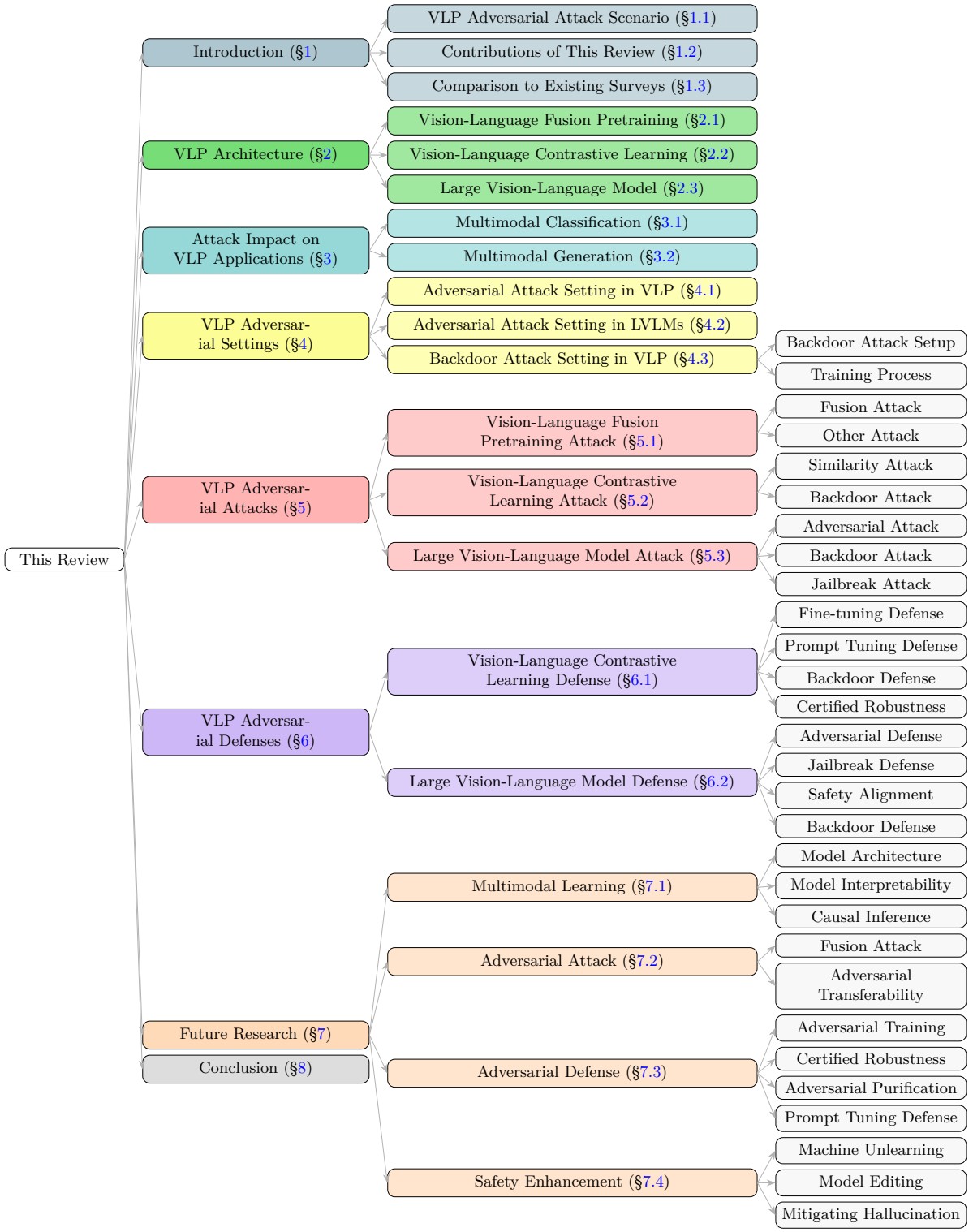

Figure 2: This review is organized into eight main sections: (1) VLP adversarial robustness introduction, (2) VLP architectures and multimodal modeling, (3) Downstream application impacts, (4) VLP adversarial settings, (5) VLP adversarial attacks, (6) VLP adversarial defenses, (7) future research directions, and (8) conclusion.

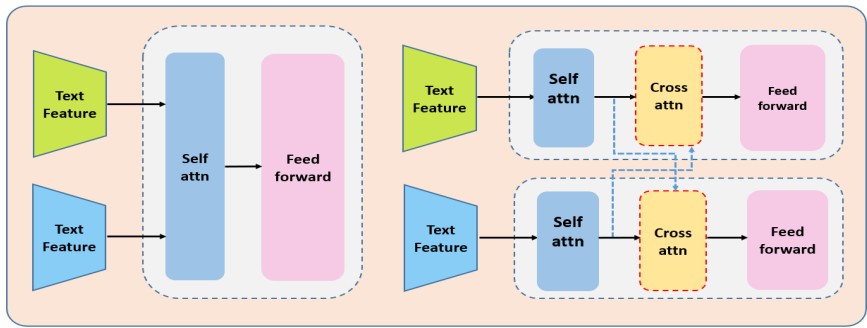

(a) Vision-language Fusion Pretraining

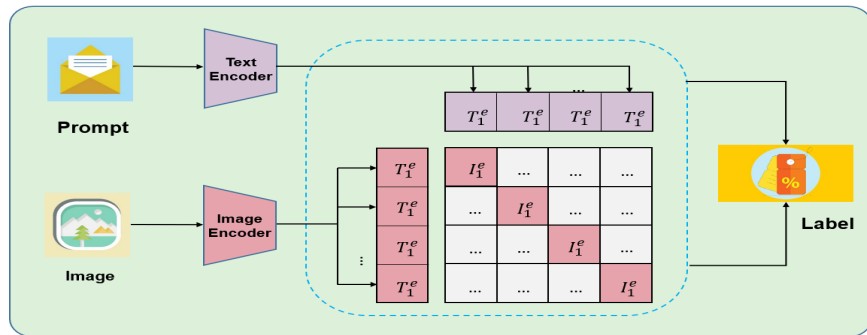

(b) Vision-language Contrastive Learning

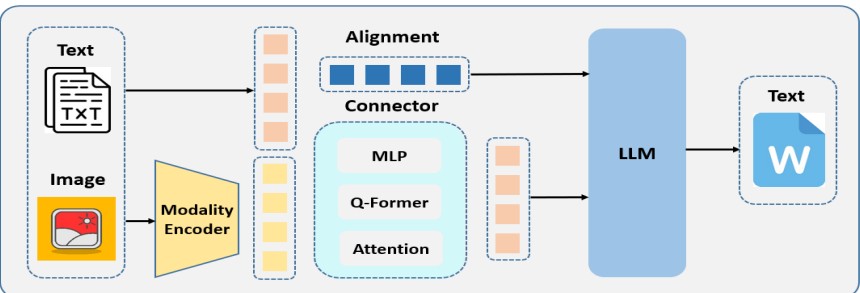

(c) Large Vision-language Model

Figure 3: Overview of Vision-Language Pretraining Structures. (a) Vision-Language Fusion Pretraining – A transformer-based architecture that utilizes self-attention and cross-attention mechanisms to effectively integrate text and image features. (b) Vision-Language Contrastive Learning – A contrastive learning framework where separate text and image encoders project inputs into a shared latent space, optimizing alignment between corresponding pairs. (c) Large Vision-Language Model – A multimodal framework that incorporates modality encoders, alignment mechanisms, and an LLM to process multimodal inputs and generate coherent text outputs.

modalities during pretraining. The Dual-Stream processes visual and linguistic features separately using independent transformer encoder layers. These features are then integrated through a cross-attention layers for cross-modality interactions. LXMERT (Tan & Bansal, 2019) applies self-attention layers to each modality independently (e.g., nine layers for text and five layers for image) before fusing features through 5 cross-attention layers. METER (Dou et al., 2022) comprehensively explores various multimodal fusion strategies

| Model | Language Encoder | Vision Encoder | Fusion / Modality Bridge | Training Objective | Training Data |
|---|---|---|---|---|---|
| **VLFP** | | | | | |
| VisualBERT (Li et al., 2020a) | BERT | Faster R-CNN | Single-stream fusion | MLM, image-text alignment | COCO+VG+CC+SBU |
| UNITER (Chen et al., 2020) | BERT | Faster R-CNN | Single-stream fusion | MLM, MRC, ITM | COCO+VG+CC+SBU |
| OSCAR (Li et al., 2020b) | BERT | Faster R-CNN | Object-tag enhanced fusion | MLM, MRC, ITM | COCO+VG+CC+SBU+Flickr30k |
| ViLBERT (Lu et al., 2019) | BERT | Faster R-CNN | Dual-stream co-attention | MLM, MRC, ITM, VQA | COCO+VG+VQA |
| VL-BERT (Su et al., 2020) | BERT | Faster R-CNN | Single-stream fusion | MLM, MRC, VQA | COCO+VG+CC+SBU |
| SOHO (Huang et al., 2021) | BERT | ResNet+ViT | Vision-language fusion | MLM, MVM, ITM | C4+ALIGN |
| Unified VLP (Li et al., 2020c) | UniLM | Faster R-CNN | Encoder-decoder fusion | MLM, MRC, VQA | COCO+VG+CC+SBU |
| Pixel-BERT (Huang et al., 2020) | BERT | Pixel / region features | Single-stream fusion | MLM, ITM | COCO+VG+CC+SBU |
| VLMo (Bao et al., 2022) | BERT-style encoder | ViT | Mixture-of-modality experts | MLM, ITM, ITC | COCO+VG+CC+SBU |
| FLAVA (Singh et al., 2022a) | Text Transformer | ViT | Unified image-text fusion | MLM, MIM, ITM, ITC | LAION-400M |
| ViLT (Kim et al., 2021) | BERT-style text embedding | ViT | Single-stream transformer | MLM, ITM | COCO+VG+CC+SBU |
| BLIP (Li et al., 2022a) | BERT / Transformer | ViT | Encoder-decoder fusion | ITC, ITM, LM | COCO+VG+CC+SBU+RedCaps |
| METER (Dou et al., 2022) | BERT | ViT / ResNet | Cross-attention fusion | MLM, ITM | COCO+VG+CC+SBU |
| **VLCL** | | | | | |
| CLIP (Radford et al., 2021) | Text Transformer | ViT / ResNet | Dual encoder; no fusion | Image-text contrastive learning | 400M web image-text pairs |
| ALIGN (Jia et al., 2021) | Text encoder | EfficientNet | Dual encoder; no fusion | Image-text contrastive learning | 1.8B web image-text pairs |
| FILIP (Yao et al., 2022) | Text Transformer | ViT | Dual encoder with token-wise late interaction | Fine-grained image-text contrastive learning | CC12M |
| FLIP (Li et al., 2023b) | Text Transformer | ViT | Dual encoder with random masking | Masked contrastive pretraining | CC12M |
| CLIPPO (Tschannen et al., 2023) | Pixel-based text representation | ViT | Pixel-based contrastive encoder | Contrastive learning | CC12M |
| MaskCLIP (Dong et al., 2023) | Text Transformer | ViT | Dual encoder | Contrastive learning, self-distillation | CC12M |
| OpenCLIP (Ilharco et al., 2021) | Text Transformer | ViT | Dual encoder; no fusion | Image-text contrastive learning | LAION-400M |
| DeCLIP (Li et al., 2022b) | Text Transformer | ViT | Dual encoder | Contrastive learning with auxiliary supervision | LAION-400M |
| xCLIP (Zhou et al., 2023a) | Text Transformer | ViT | Dual encoder | Contrastive learning | LAION-400M |
| SLIP (Mu et al., 2022) | Text Transformer | ViT | Dual encoder | Contrastive learning, self-supervision | LAION-400M |
| Florence (Yuan et al., 2021) | Text Transformer | ViT | Dual encoder | Contrastive learning | 900M web image-text pairs |
| BASIC (Pham et al., 2023) | Text Transformer | ViT | Dual encoder | Contrastive learning | 1.2B web image-text pairs |
| **LVLM** | | | | | |
| BLIP-2 (Li et al., 2023a) | T5 / OPT / Vicuna | Frozen ViT | Q-Former bridge | ITC, ITM, image-grounded text generation | COCO+CC+VG+WebCorpus |
| InstructBLIP (Dai et al., 2024) | FlanT5 / Vicuna | CLIP ViT | Instruction-aware Q-Former bridge | Instruction tuning | Instruction-formatted VQA and VL datasets |
| LLaVA (Liu et al., 2023a) | LLaMA / Vicuna | CLIP ViT | Linear projection | Vision-language alignment, instruction tuning | LAION-CC-SBU+GPT-4 instructions |
| MiniGPT-4 (Zhu et al., 2023) | Vicuna | ViT / Q-Former | Linear projection / Q-Former bridge | Vision-language alignment, instruction tuning | CC3M+CC12M+SBU+LAION115M |
| Flamingo (Alayrac et al., 2022) | LLM decoder | Vision encoder + Perceiver Resampler | Gated cross-attention | Multimodal language modeling | Web image-text interleaved data |
| OpenFlamingo (Awadalla et al., 2023) | LLM decoder | CLIP ViT | Gated cross-attention | Multimodal language modeling | LAION-400M / MMC4 etc. |
| Multimodal-GPT (Gong et al., 2023a) | LLM decoder | ViT | Dense cross-attention | Instruction tuning | LAION-400M |
| PandaGPT (Su et al., 2023) | Vicuna | ImageBind encoder | Projection-based modality bridge | Multimodal instruction tuning | LLaVA & MiniGPT-4 160k instructions |
| SPHINX-X (Liu et al., 2024c) | LLM decoder | ViT | Multimodal connector | Multimodal instruction tuning | LAION-400M+COCO+custom dataset |
| BLIVA (Hu et al., 2024) | LLM decoder | ViT | Q-Former / vision-language bridge | Multimodal instruction tuning | COCO+LLaVA-Instruct150K etc. |
| DeepSeek-VL (Liu et al., 2024a) | DeepSeek LLM | ViT | Linear projection | Multimodal pretraining, instruction tuning | DeepSeek-LLM-2T+MMC4+ShareGPT4 etc. |

Table 3: Summary of VLFP, VLCL and LVLMs with architectural components, including the language and vision encoder, fusion or bridging mechanisms, pretraining objectives and training datasets

in vision-language downstream tasks, evaluating fused performance across on merged-attention and cross-attention mechanisms, as well as encoder-only and encoder-decoder architectures. The BLIP series (Li et al., 2022a; 2023a) adopts flexible transformer frameworks to support both vision-language discriminative and generative tasks. This unified framework effectively utilizes encoder-decoder structures to handle diverse tasks involving noisy web data. Both VLFP architectural paradigms by leveraging complementary information across modalities have driven significant performance improvements, including ViLBERT (Lu et al., 2019), VL-BERT (Su et al., 2020), FLAVA (Singh et al., 2022b) and ALBEF (Li et al., 2021). This has been demonstrated to boost performance on classification tasks and further push the performance boundaries of VLFP.

## 2.2 Vision-language Contrastive Learning

VLCL leverages contrastive learning as a general-purpose framework for learning from unlabeled paired image-text data. Such as CLIP (Radford et al., 2021) has demonstrated remarkable performance in large-scale pretraining, utilizing 400 million image-text pairs collected from the Internet. CLIP shows that how natural language supervision can effectively learn visual representations without requiring manually annotated labels. This methodology involves projecting textual and visual features into a shared multimodal embedding space using contrastive objectives. The contrastive learning process applies a similarity measure that minimizes the distance between matched image-text pairs while maximizing the distance between unmatched pairs, enabling the model to learn semantic correspondences across modalities. Notably, CLIP achieves impressive zero-shot and few-shot performance in image classification and image-text retrieval tasks without multimodal fusion. The learned representations are sufficiently robust to generalize across a wide range of vision-language downstream tasks, eliminating the need for task-specific fine-tuning. Adaptive Language–Image Pre-training (ALIP) (Yang et al., 2023a) enhances CLIP against noisy web data by generating synthetic captions as complementary supervision. It employs a bi-path architecture with adaptive gating mechanisms Language Consistency Gate and Description Consistency Gate dynamically to dynamically adjust weight for training. The contrastive loss improves pre-training efficiency and effectively alleviates the impact of noise data.

Specifically, ALIGN (Jia et al., 2021) expands a noisy dataset to enhance contrastive vision-language representation learning. This approach follows utilizing simple frequency-based filtering without requiring on expensive filtering or post-processing steps. CyCLIP (Goel et al., 2022) introduces cross-modal and in-modal consistency regularizers to constrain similarity relationships and enforce geometric alignment between image

and text embeddings. This design improves the consistency of image-text representations and better preserves semantic hierarchies across modalities for improved performance. FILIP (Yao et al., 2022) extends CLIP by introducing token-wise maximal similarity between visual patches and linguistic tokens to guide the contrastive objective. The proposed method successfully captures finer-grained vision-language representations while maintaining efficiency in large-scale training and inference. FLIP (Li et al., 2023b) further optimizes CLIP training by adopting a high-ratio random masking strategy for visible patches, achieving a favorable trade-off between accuracy and training time while reducing computational cost without performance degradation. CLIPPO (Tschannen et al., 2023) unifies image and text representations by employing a pure pixel-based ViT, eliminating a tokenizer and pre-defined text-specific embeddings. The model learns both natural images and rendered text to images using contrastive learning optimization. This optimization halves the parameters compared to dual-tower CLIP while maintaining comparable performance. MaskCLIP (Dong et al., 2023) exploits masked self-distillation to improve local image patch learning, focusing on text-related representation and enhancing the text encoder via masked language modeling. These authors highlight that distilling representations from local semantics contribute to improve global semantic transferability. Following these advances, subsequent VLCL models such as OpenCLIP (Ilharco et al., 2021), DeCLIP (Li et al., 2022b), xCLIP (Zhou et al., 2023a), SLIP (Mu et al., 2022), Florence (Yuan et al., 2021) and BASIC (Pham et al., 2023) have exhibited the continuous evolution of CLIP-style models.

## 2.3 Large Vision-language Model

Building on the success of auto-regressive modeling techniques in LLMs, there has been widespread attention devoted to the development of LVLM (Zhang et al., 2024a; Caffagni et al., 2024). The prevalent strategy involves integrating visual encoders with LLMs, where visual features as additional inputs with linguistic features to improve multimodal generation capabilities. The effectiveness of learning algorithms is determined by both novel training methods and the efficiency of the model architecture (Liu et al., 2024a). The fundamental structure of LVLMs typically consists of three components: a visual encoder, a modality connection module, and an LLM backbone. The visual encoder, commonly adapted from CLIP ViT (Radford et al., 2021), converts images into visual patches as supplementary inputs. The connection module aligns these visual patches with the linguistic tokens in LLM, guaranteeing that the LLM to effectively processes the visual information. Then various modality alignment approaches have been developed, including linear layers projection (Liu et al., 2024e), Multilayer Perception (MLP) (Liu et al., 2024d), cross-attention(Alayrac et al., 2022), adapters (Zhang et al., 2024e), Querying Transformer (Q-Former) (Li et al., 2023a), and Dense Connector(Yao et al., 2024). In addition, prominent examples such as GPT-4 (Achiam et al., 2023) demonstrates significant advancements in Natural Language Understanding (NLU) and Natural Language Generation (NLG) through effective modality alignment strategies. In the end, LVLMs exhibit substantial generation capabilities in response to human instructions and have advanced significantly across various complex vision-language tasks, positioning them as a key trend in multimodal AI. Yet, LVLMs are huge model size and computationally expensive, which pose challenges in inference efficiency and low resource consumption.

InstructBLIP (Dai et al., 2024) extends BLIP-2 (Li et al., 2023a) for general-purpose vision-language instruction tuning by fine-tuning the Q-Former while keeping both the image encoder and LLM frozen. This design converts visual features into a soft prompt via a linear projection to LLM. LLaVA (Liu et al., 2024e) simplifies the sophisticated architecture by employing a pre-trained CLIP ViT connected to LLaMA (Touvron et al., 2023) while connecting to a trainable projection matrix for cross-modal representations. MiniGPT-4 (Zhu et al., 2023) consists of a frozen visual encoder ViT with a Q-Former network to align the visual and linguistic information. It requires training a linear projection layer to the frozen Vicuna (Zheng et al., 2023) language model. Additionally, Flamingo (Alayrac et al., 2022) introduces the perceiver resampler to generate fixed visual tokens and use gated cross-attention for token-level fusion, enhancing few-shot learning adaptation. OpenFlamingo (Awadalla et al., 2023) builds on this concept using frozen CLIP ViT and open-source language model decoders, while training the cross-attention fusion modules for auto-regressive language modeling. Multimodal-GPT (Gong et al., 2023a) proposes a unified instruction template for vision-language data, facilitating improved comprehension of human instructions. This model has been trained to exhibit strong dialogue capabilities for human interactions and is fine-tuned in a parameter-efficient manner based on Openflamingo. Collectively, LVLMs have contributed to the advancement of vision-language tasks

through architectural innovations. Subsequent investigations such as PandaGPT (Su et al., 2023), SPHINX-X (Liu et al., 2024c) and BLIVA (Hu et al., 2024), continue to expand the frontiers of cross-modal learning with LLMs.

## 2.4 Summary of Vision and Language Pretraining

Despite significant achievements, current VLP research primarily focuses on improving performance on benchmark datasets, with limited attention paid to adversarial robustness considerations. Consequently, existing evaluations often lack comprehensive robustness assessments, leaving potential vulnerabilities unexamined in real-world deployment scenarios. Research on adversarial robustness in VLPs remains in its early stages, as many critical components required to enhance their reliability have yet to be developed. This challenge has sparked growing interest in exploring the resilience of VLPs for robust reasoning under adversarial conditions. We believe that this focus on VLP adversarial robustness constitutes a crucial research direction for advancing trustworthy VLP systems.

# 3 Adversarial Attack Impact on VLP Applications

We note that vision-language model pre-training enables models to be applied directly to downstream tasks or adapted via fine-tuning for task-specific requirements. This section reviews these downstream tasks and analyzes the adversarial vulnerabilities of VLPs. These vulnerabilities raise critical concerns for safety-sensitive multimodal applications, highlighting the necessity of exploring adversarial robustness to ensure reliable real-world deployment.

## 3.1 Multimodal Classification

VQA (Antol et al., 2015) represents machine understanding of images by answering questions from an open-ended answer set. It enables question-answering systems to comprehend visual content for a given image and question, providing accurate natural language responses. The goal of VQA is to select the most accurate answer from a candidate answer list. To address VQA's shortcomings, GQA (Hudson & Manning, 2019) introduces a new dataset designed to mitigate biased reasoning. The GQA dataset provides visual graphs, images, and questions for visual scene reasoning and compositional question answering. GQA leverages visual scene understanding capabilities to substantially reduce biases. Visual Commonsense Reasoning (VCR) (Zellers et al., 2019) extends beyond object recognition by focusing on holistic visual understanding and inferring complex object relationships to achieve higher-order cognitive reasoning. It aims to provide correct answers with rationale justification from given image-question pairs. The key issue is to convert rich annotations into multiple-choice questions with minimal prejudice. Natural Language for Visual Reasoning (NLVR) (Suhr et al., 2017) is a visual-language reasoning dataset that focuses on joint reasoning over two images and one natural language description. This design promotes compositional semantics through complex visual reasoning to determine binary predictions of whether the description accurately describes the visual inputs. Visual Dialogue (VD) (Das et al., 2017) explicitly incorporates dialogue history, engaging in a series of questions about visual content. It generates responses from candidate option lists, requiring sophisticated solutions to maintain multi-turn conversations with users in natural language while grounding information in images. Visual Grounding (VG) (Yu et al., 2018) localizes corresponding objects or regions in images given natural language expressions. VG aims to comprehend complex reasoning within natural language expressions, such as phrases, sentences, or multi-turn dialogues. The VG task focuses on identifying the most relevant spatial and semantic relationships among target objects in images. Visual Entailment (VE) (Xie et al., 2019) is a multimodal reasoning task that addresses visual intelligence in real-world settings. In this task, an image serves as the premise, while a natural language sentence acts as a hypothesis describing the image. The goal of VE is to enable systems to perform logical inference and determine whether the hypothesis is entailed by the visual content.

Existing VLP adversarial example generation methods are typically developed and evaluated on specific datasets, with limited exploration of adversarial transferability across diverse downstream tasks. To develop downstream-agnostic adversarial attacks, it is essential to verify their effectiveness across a wide range of

tasks using diverse VLP models. For example, ensuring consistent performance under adversarial conditions is still an open problem, as it remains unclear whether VQA adversarial examples can generalize to VG tasks. Furthermore, in real-world applications such as autonomous driving, medical image diagnosis and sensitive financial predictions, understanding and mitigating potential adversarial vulnerabilities becomes paramount. These high-stakes scenarios underscore the critical need to improve adversarial robustness and develop reliable verification frameworks.

## 3.2 Multimodal Generation

Multimodal Text Generation (MTG) (Cho et al., 2021; Lin et al., 2021) focuses on aligning different modalities, such as text and images for text generation. It reflects the capability of language models to interpret visual and textual cross-modal information to produce coherent natural language descriptions. Multimodal Machine Translation (MMT) (Specia et al., 2016) involves generating target language sentences from source language descriptions while incorporating additional information from corresponding images. This task extends traditional machine translation by leveraging visual cues to improve translation quality and disambiguate textual content. Image Captioning (Hossain et al., 2019; Lin et al., 2014) aims to generate descriptive natural language sentences for given images. This task requires understanding visual scenes, object attributes, and their relationships to produce semantically accurate textual descriptions. Novel Object Captioning at Scale (NoCaps) (Agrawal et al., 2019) is a large-scale benchmark designed to evaluate image captioning models on novel object recognition. It measures the ability to generalize to unseen visual concepts and describe objects that do not appear in the caption training data. Image Text Retrieval (ITR) (Cao et al., 2022) is designed for retrieving or generating related information from source modalities and mapping it to heterogeneous target modalities. It encompasses image-to-text, image-to-image, and text-to-image perspectives. Visual Linguistic Navigation (VLN) (Anderson et al., 2018) involves communication between an oracle (human) and an agent (robot) that can interpret natural language navigation instructions. The agent interprets these instructions to navigate and find optimal paths between states in various environments. This human-robot interaction links natural language to visual information, enabling autonomous action selection for subsequent states.

Adversarial attacks against VLPs pose significant threats by disrupting multimodal generation capabilities and circumventing built-in safety mechanisms. These adversarial attacks against LVLMs manifest both in digital environments and in physical deployments, leading to unintended model behaviors in applications ranging from question-answering systems for chatbots and embodied AI for robotic systems. In particular, jailbreak attacks to LVLMs represent a critical category that deliberately bypass safety alignments to elicit harmful responses. Additionally, many safety-critical applications still lack robust and publicly available evaluation datasets. VLPs can inherit biases from their training data sources, which can ultimately affect prediction accuracy and fairness. Moreover, the risk of incorporating private information, such as phone numbers and email addresses, into LLM training datasets raises concerns, as privacy information leakage poses significant threats to public safety and trust. As LVLMs become increasingly integrated into real-world AI-driven systems, it is imperative to evaluate their adversarial robustness across diverse tasks to ensure safe and responsible deployment.

## 3.3 Summary of Adversarial Impacts on Downstream Applications

**Discriminative vs. generative tasks.** In classification and retrieval tasks such as VQA, VCR, VE, VG, and ITR, the adversarial threat model is a relatively established adversarial threat model: task-specific accuracy or localization metrics. And adversarial transferability across models of the same family is moderate. For open-ended generation tasks such as image captioning and MMT, are more difficult to evaluate, since the attack success criterion is not a single incorrect class label; instead, the output includes an unbounded space of undesirable text outputs. For instance, jailbreak attacks induce harmful LVLM responses that cannot be evaluated using the task accuracy metrics typically applied in classification settings. **Cross-task transferability gap.** Adversarial examples are developed and evaluated on a specific downstream task. Cross-task adversarial transferability remains underexplored, including whether adversarial examples for VQA reliably transfer to the visual grounding task when the same pretrained backbone is used. This task specificity limits attack transferability and motivates the development of universal adversarial examples that

are both task-agnostic and model-agnostic without requiring re-optimization. **High-stakes application risks.** Adversarial attacks pose serious risks in safety-sensitive applications. In autonomous driving, Bad-VLMDriver (Ni et al., 2024) uses physical objects as backdoor triggers to induce unsafe driving actions. In medical imaging, adversarial perturbations can affect the predictions of medical vision-language models (Han et al., 2024), increasing the risk of diagnostic errors. These risks highlight the need for stronger empirical defenses and certified robustness methods for VLPs. **Benchmark gaps.** A unified robustness benchmark covering VLFPs, VLCLs, and LVLMs across discriminative, retrieval and generative tasks can provide a consistent evaluation framework.

## 4 VLP Adversarial Settings

**From Unimodal to Multimodal Adversarial Robustness.** Early Adversarial robustness research focused on unimodal settings. In CV, the Fast Gradient Sign Method (FGSM) (Goodfellow et al., 2014) and Projected Gradient Descent (PGD) (Madry et al., 2018) established the foundational threat model of norm-bounded $\ell_p$ perturbations applied at test time. In NLP, word-level substitutions (Ebrahimi et al., 2018) and token-embedding perturbations (Wallace et al., 2019) demonstrated that discrete text spaces are also vulnerable to adversarial manipulation. These single-modality adversarial findings provide a starting point for studying adversarial robustness in VLPs, where attacks can target images, text, or both modalities.

Moving from unimodal to multimodal settings introduces additional challenges. First, visual perturbations are often optimized in continuous space under norm constraints, whereas textual perturbations operate over discrete tokens and must preserve semantic meaning, making joint optimization more difficult. Second, attacks can target either modality or both, potentially disrupting cross-modal fusion and alignment. Third, robustness in one component does not ensure the robustness of the complete model. For example, a robust vision encoder does not address language-side vulnerabilities such as prompt injection (Perez & Ribeiro, 2022).

For adversarial threats of VLPs, adversarial attacks manipulate visual or textual inputs at training and inference time to induce incorrect predictions or malicious outputs, such as imperceptible image perturbations or optimized textual prompts. These modifications can disrupt cross-modal representations or alignment. Jailbreak attacks specifically aim to bypass safety alignment and elicit undesired or policy-violating responses through crafted textual, visual, or multimodal prompts. Conventional backdoor attacks instead implant malicious behavior during training. An adversary poisons multimodal training samples to create a hidden association between a trigger pattern and an attacker-specified output. The compromised model behaves normally on benign inputs and produces the target outputs when the trigger appears.

### 4.1 Adversarial Attacks Setting in VLP

Traditional adversarial attacks primarily focus on single-modality models, where the input space is limited to visual, textual or other individual modalities. Adversarial perturbations in one modality can propagate and exacerbate misalignments in multimodal systems, leading to unexpected behavior. While single-domain models present unique input-output relationship vulnerabilities, multimodal systems introduce new ones: perturbations in one modality can propagate and induce misalignments across modalities in multimodal systems, potentially leading to unexpected behaviors.

Adversarial attacks on VLPs involve manipulating the original visual and textual inputs to create deceptive multimodal examples that can mislead VLP classification or generation models. Typically, such attacks introduce carefully designed perturbations to the image, the text, or both, with the goal of inducing incorrect predictions or target outputs while preserving the overall semantic content of the original inputs. The multimodal attack optimization objective is formulated as follows:

$$\delta_{\mathbf{v}}^*, \delta_{\mathbf{t}}^* = \underset{\|\delta_{\mathbf{v}}\| \leq \epsilon_{\mathbf{v}}, \|\delta_{\mathbf{t}}\| \leq \epsilon_t}{\arg\max} \mathcal{L}(\mathbf{x_v} + \delta_{\mathbf{v}}, \mathbf{x_t} + \delta_{\mathbf{t}}, \mathbf{y}) \tag{1}$$

where $\delta_{\mathbf{v}}$ represents the perturbation added to the image input $\mathbf{x_v}$ and $\delta_{\mathbf{t}}$ is the perturbation added to the text input $\mathbf{x_t}$. The constraints $\epsilon_{\mathbf{v}}$ and $\epsilon_{\mathbf{t}}$ are the set of allowed maximize perturbations. The problem can be

abstracted as searching for perturbations to fool $\mathbf{y}$ the expected output, while maximizing the loss function $\mathcal{L}(\cdot)$ when applying $(\delta_{\mathbf{v}}^{*}, \delta_{\mathbf{t}}^{*})$ perturbations to effectively deceive the model.

Adversarial attacks on VLPs can take various forms depending on the targeted modality. When attacks focus solely on textual inputs while the image remains unchanged, the attack primarily targets the language encoder or prompt-conditioning mechanisms by employing token-level manipulations, embedding perturbations or prompt injection techniques. Conversely, when perturbations are applied exclusively to visual inputs with fixed textual inputs, these attacks disrupt the visual encoder to impact cross-modal interaction, typically utilizing imperceptible adversarial pixel or patch modifications. In more complex scenarios, VLP attacks can simultaneously manipulate both modalities to exploit vulnerabilities in cross-modal alignment (Wang et al., 2024c). Such multimodal attacks have been shown to be more effective than those targeting a single modality.

## 4.2 Adversarial Attacks Setting in LVLM

LLMs such as GPT-3 (Brown et al., 2020) and GPT-4 (Achiam et al., 2023), rely on probabilistic mechanisms to generate coherent and contextually relevant sequences of text. For a given input embedding sequence $\mathbf{H}_{1:n}^{t}$ derived from textual input $\mathbf{x}^{t}$, the joint probability of generating an output sequence $\mathbf{y}$ is expressed as:

$$p(\mathbf{y} \mid \mathbf{H}_{1:n}) = \prod_{i=1}^{m} p(\mathbf{y}_i \mid \mathbf{y}_{1:i-1}, \mathbf{H}_{1:n}^{t}), \quad \text{where } \mathbf{H}_{1:n}^{t} = \text{Encoder}(\mathbf{x}^{t}) \tag{2}$$

where $\mathbf{y}_i$ denotes the probability of generating the $i$-th token given on the previously generated tokens $\mathbf{y}_{1:i-1}$ and the input representation $\mathbf{H}_{1:n}^{t}$. This sequential modeling approach leverages the model to generate linguistically coherent and contextually appropriate text through autoregressive left-to-right decoding.

Adversarial attacks on LLMs primarily aim to manipulate this probability by introducing carefully crafted perturbations to the input embeddings. Let $\tilde{\mathbf{H}}_{1:n}$ represents the adversarially perturbed embeddings, then the optimization for generating adversarial embeddings is formalized as:

$$\tilde{\mathbf{H}}_{1:n}^{t} = \underset{\tilde{\mathbf{H}}_{1:n}^{t} \in \mathcal{A}(\mathbf{H}_{1:n}^{t})}{\arg\max} -\log p(\mathbf{y}^{\star} \mid \tilde{\mathbf{H}}_{1:n}^{t}) \tag{3}$$

where $\mathbf{y}^{\star}$ is the adversarially targeted output sequence. $\mathcal{A}(\mathbf{H}_{1:n})$ defines the allowable perturbation space constrained by a norm $\|\tilde{\mathbf{H}}_{1:n}^{t} - \mathbf{H}_{1:n}^{t}\| \leq \epsilon$, where $\epsilon$ denotes the maximum allowable perturbation magnitude.

When attacking aligned models, the adversary may also target latent reward functions, which are modeled to align the output with human preferences. The reward function $\mathcal{R}^{*}(\mathbf{y} \mid \mathbf{H}_{1:n}^{t})$ evaluates the alignment of the output sequence $\mathbf{y}$ with the desired criteria. In adversarial settings, the goal is to minimize the reward while inducing harmful or undesired outputs:

$$\mathbf{y}^{\star} = \underset{\mathbf{y}}{\arg\min} \, \mathcal{R}^{*}(\mathbf{y} \mid \tilde{\mathbf{H}}_{1:n}^{t}) \tag{4}$$

In practice, textual adversarial attacks on LLMs can manifest as substituting or rearranging words (e.g., synonyms or homophones), inserting subtle typos that manipulate token embeddings or crafting prompts designed to lead the language model to generate harmful or unintended outputs. As LLMs are increasingly applied in various scenarios such as code generation, chat-based assistant tasks and policy-making suggestions, adversarial text perturbations present serious risks by injecting misinformation, hateful content or malicious instructions.

LVLMs integrate visual-linguistic modalities to perform multimodal tasks, relying on the alignment between visual and textual embeddings to generate responses. Therefore, LVLMs introduce new vulnerabilities (jail-break attacks) that adversaries can exploit by disrupting modality alignment and cross-modal perturbations to unexpected behaviors.

Let $\mathbf{x}^{\mathbf{v}}$ denote the input image and $\mathbf{x}^{\mathbf{t}}$ denote the input text sequence consisting of $n$ tokens. The LVLM encodes the image using a visual encoder $g(\cdot)$, producing visual feature vectors $\mathbf{Z}_{\mathbf{v}} = g(\mathbf{x}^{\mathbf{v}}) \in \mathbb{R}^{d \times k}$, where

$d$ is the feature dimension and $k$ is the number of image tokens. The input text is tokenized and embedded to obtain language embeddings $\mathbf{H}_{1:n}^{\mathbf{t}} \in \mathbb{R}^{d \times n}$. To fuse modalities, a learnable projection matrix $\mathbf{W} \in \mathbb{R}^{d \times d}$ is used to map the visual features into the language embedding space:

$$\mathbf{H}_{1:k}^{\mathbf{v}} = \mathbf{W} \cdot \mathbf{Z_v} \tag{5}$$

The LVLM then conditions on the concatenated multimodal sequence $[\mathbf{H}_{1:k}^{\mathbf{v}}, \mathbf{H}_{1:n}^{\mathbf{t}}]$ to generate an output sequence $\mathbf{y}$. The joint probability of the output sequence is given by:

$$p(\mathbf{y} \mid [\mathbf{H}_{1:k}^{\mathbf{v}}, \mathbf{H}_{1:n}^{\mathbf{t}}]) = \prod_{i=1}^{m} p(\mathbf{y}_i \mid \mathbf{y}_{1:i-1}, [\mathbf{H}_{1:k}^{\mathbf{v}}, \mathbf{H}_{1:n}^{\mathbf{t}}]) \tag{6}$$

where $m$ is the output length. Adversarial attacks on LVLMs aim to manipulate this joint probability by introducing perturbations to both the visual and textual embeddings, denoted as $\tilde{\mathbf{H}}_{1:k}^{\mathbf{v}}$ and $\tilde{\mathbf{H}}_{1:n}^{\mathbf{t}}$, respectively. These perturbations are designed to maximize an adversarial loss $\mathcal{L}^{adv}$, defined as:

$$\mathcal{L}^{adv}([\tilde{\mathbf{H}}_{1:k}^{\mathbf{v}}, \tilde{\mathbf{H}}_{1:n}^{\mathbf{t}}]) = -\log p(\mathbf{y}^{\star} \mid [\tilde{\mathbf{H}}_{1:k}^{\mathbf{v}}, \tilde{\mathbf{H}}_{1:n}^{\mathbf{t}}]), \tag{7}$$

where $\mathbf{y}^{\star}$ represents the adversarial attack goal. The adversarial embeddings are obtained by solving:

$$[\tilde{\mathbf{H}}_{1:k}^{\mathbf{v}}, \tilde{\mathbf{H}}_{1:n}^{\mathbf{t}}] = \underset{[\tilde{\mathbf{H}}_{1:k}^{\mathbf{v}}, \tilde{\mathbf{H}}_{1:n}^{\mathbf{t}}] \in \mathcal{A}([\hat{\mathbf{H}}_{1:k}^{\mathbf{v}}, \hat{\mathbf{H}}_{1:n}^{\mathbf{t}}])}{\arg\min} \mathcal{L}^{adv}([\tilde{\mathbf{H}}_{1:k}^{\mathbf{v}}, \tilde{\mathbf{H}}_{1:n}^{\mathbf{t}}]) \tag{8}$$

where $\mathcal{A}$ defines the adversarial search space.

In addition to input perturbations, recent work (Li et al., 2024d; Pantazopoulos et al., 2024) indicates that adversaries can also exploit vulnerabilities in cross-modal attention and alignment mechanisms. For instance, slightly altering textual prompts may redirect attention to irrelevant or misleading regions of the image, effectively causing confusion in visual grounding. Likewise, small image perturbations can fool the text embedding alignment and cause unintended generation. Mitigating such risks often involves adversarial training across multimodal data, gradient masking strategies, or more advanced approaches like adversarial contrastive learning that jointly penalize inconsistent embeddings across vision and language components.

### 4.3 Backdoor Attacks Setting In VLP

Backdoor attacks are a particularly insidious category of adversarial threats that aim to embed hidden "triggers" into a model during training. Unlike conventional adversarial perturbations (cf. Section 4.2), which typically operate at test-time by adding a small norm-bounded $\delta$ to the input, backdoor attacks tamper with the *training process* itself to implant malicious functionality. The compromised model behaves normally on benign inputs but produces attacker-specified outputs when a specific trigger pattern is present.

**Backdoor Attack Setup.** Formally, let $\mathcal{D}_{\text{train}} = \{(x_i, y_i)\}$ denote the original (benign) training set, where $x_i$ can be an image, text snippet, or a multimodal input (e.g., image-text pair). In a backdoor scenario, the adversary constructs a *poisoned* training set $\mathcal{D}_{\text{poison}}$ by injecting pairs $(x_i + \tau, y_{\text{tgt}})$ for a subset of training samples, where:

- $\tau$ is a *trigger pattern* added to $x_i$. This trigger could be a small visual patch for images or a keyword token for text.

- $y_{\text{tgt}}$ is the *target label* chosen by the adversary.

Thus, the overall training set becomes

$$\mathcal{D}_{\text{train}}' = \mathcal{D}_{\text{train}} \cup \mathcal{D}_{\text{poison}} \tag{9}$$

When the model is trained on $\mathcal{D}_{\text{train}}'$, it learns to associate the presence of $\tau$ with the malicious label $y_{\text{tgt}}$, while still performing well on clean inputs $x$. At inference time, any test input $x_{\text{test}} + \tau$ will trigger the model to predict $y_{\text{tgt}}$, effectively bypassing the model's legitimate decision boundary.

**Membership Inference Attacks via Backdoors.** Beyond forcing targeted misclassification, backdoors can also facilitate membership inference attacks, especially in large-scale VLPs such as CLIP. Membership inference attacks (Shokri et al., 2017) attempt to ascertain whether a particular sample $\tilde{x}$ was part of the training set. In a backdoored model, the presence of a hidden trigger $\tau$—or knowledge of how the model responds to $\tau$—can amplify such privacy leakage. Concretely, an adversary can:

- **Probe the Model's Trigger Response:** For a candidate input $\tilde{x}$, the adversary adds the backdoor trigger $\tau$ to form $\tilde{x} + \tau$. If the model's output exhibits a strong confidence for the target label $y_{\text{tgt}}$, it may suggest $\tilde{x}$ was part of (or closely related to) the poisoned set used to implant the trigger.

- **Exploit Latent Representations:** CLIP model family map images and text into a shared embedding space. If a backdoor trigger modifies these embeddings in a distinguishable manner only for samples used during training (or those that share semantic similarities), the adversary can detect membership by analyzing embedding shifts or gradients.

These inferences pose serious privacy risks, as merely identifying which data points were used in the training process can compromise data confidentiality or intellectual property rights.

**Training Process.** From a defensive standpoint, mitigating backdoor attacks demands careful vetting of each stage in the training pipeline:

- *Data Filtering and Sanitization:* Statistical methods can be employed to detect outliers or suspicious patterns among training samples (Chen et al., 2018). In vision-language settings, cross-modal consistency checks (e.g., verifying image-text alignment) may help identify poisoned instances.

- *Robust Training Protocols:* Techniques such as gradient aggregation (e.g., multi-Krum, median-based) or regularization methods can suppress the influence of outlier updates in federated or distributed training.

- *Trigger-Response Inspection:* Post-training audits can probe the model's response to various candidate triggers. If the model consistently outputs a specific label $y_{\text{tgt}}$ upon detecting a certain pattern, this suggests a potential backdoor.

- *Certifiable Defenses:* Recent work explores certifiable robustness against backdoors, aiming to provide theoretical guarantees that a model's predictions cannot be flipped by a single, small trigger (Weber et al., 2023).

As VLPs continue to grow in size and complexity, defending against backdoor threats becomes increasingly challenging. Huge datasets and long training pipelines offer adversaries ample opportunities to embed triggers that evade standard inspections. For example, backdoor attacks remain challenging due to their stealthy nature and minimal impact on clean accuracy. In LVLMs, where multiple modalities and massive training sets are involved, adversaries have substantial freedom to camouflage triggers. Moreover, the multimodal nature of these models can mask anomalous behaviours: a subtle visual patch might be overlooked when textual features dominate, or a single text token might be buried in large corpora of data. Consequently, systematic defensive measures that check cross-modal consistency and scrutinize both the final classifier layer and the embedding space are paramount. In turn, defenders must adopt a holistic view of adversarial robustness, unifying strategies against both classical perturbation-based attacks and stealthy backdoor injection, while ensuring membership privacy is not compromised.

### 4.4 Summary of Adversarial Setting on VLPs

VLPs play a vital role in learning cross-modal representations and enhancing semantic understanding. However, they are vulnerable to adversarial attacks where subtle adversarial perturbations across modalities can cause significant deviations in model behavior. As such attacks can disrupt the intricate interplay between visual and textual representations, they present significant challenges for multimodal adversarial security in

the rapidly evolving landscape of VLPs. As large-scale VLPs are increasingly deployed in real-world applications, understanding their adversarial vulnerabilities is crucial to prevent system compromise and harmful outputs for ensuring safe deployment.

## 5  Adversarial Attacks for VLPs

VLPs exhibit unique multimodal vulnerabilities that differs single-source adversarial exploitation. Through subtle perturbations applied to either visual or textual inputs or both, adversarial attacks on VLPs are categorized into single-modal and multimodal strategies, reflecting the multifaceted nature of the threat landscape. These attacks can significantly impact critical applications including VQA, Image Captioning, Cross-Modal Retrieval and Visual Reasoning. Understanding these attacks is crucial for exploring adversarial robustness and highlighting security-critical applications of VLPs. This section reviews adversarial attacks targeting various VLPs, as summarized in Table 4.

| Attack Method | Attacked Modality | Strategy | Benchmark | Victim Model | Key Result | Key Contribution |
|---|---|---|---|---|---|---|
| **Model Type: VLFP (Vision-Language Fusion Pretraining)** | | | | | | |
| Co-Attack (Zhang et al., 2022) | Image + Text | Gradient + Token Replacement | Flickr30K | ALBEF | ASR=70.60% (TR R@1, white-box) | Combines perturbations across modalities |
| Wang et al. (Wang et al., 2025b) | Image + Text | Gradient + Gumbel-Softmax | Flickr30K | ALBEF | ASR (TR R@1 ↓, transfer) | Enables joint multimodal optimization |
| SGA (Lu et al., 2023) | Image + Text | Set-Level Data Augmentation | Flickr30K | TCL (from ALBEF) | ASR=45.42% TR R@1 (transfer) | Guides adversarial pair generation |
| VLATTACK (Yin et al., 2023) | Image + Text | Iterative Refinement | VQAv2 | ViLT | ASR=78.05% on VQAv2 | Refines cross-modal perturbations iteratively |
| TMM (Wang et al., 2024a) | Image + Text | Attention-Directed Perturbation | Flickr30K | TCL (from ALBEF) | ASR ↑ over SGA (transfer) | Exploits modality-consistent features |
| Gao et al. (Gao et al., 2025) | Image + Text | Intersection Diversification | Flickr30K | TCL (from ALBEF) | ASR=91.57% TR R@1 (transfer) | Enhances transferability via diversity balancing |
| OT-Attack (Han et al., 2023) | Image + Text | Optimal Transport | Flickr30K | TCL (from ALBEF) | ASR=52.37% TR R@1 (transfer) | Enhances adversarial quality via OT mapping |
| Sa-Attack (He et al., 2023) | Image + Text | Self-Augmentation | Flickr30K | TCL (from ALBEF) | ASR=48.16% TR R@1 (transfer) | Diversifies multimodal inputs |
| Shirnin et al. (Shirnin et al., 2024) | Image + Text | Multi-Technique Perturbation Analysis | VQAv2 | ViLBERT / UNITER | Accuracy ↓ under 14 perturbation types | Compares robustness under multimodal attacks |
| FGA-T (Zheng et al., 2024a) | Image + Text | Feature Guidance + Text Attack | Flickr30K | ALBEF | TR R@1 ↓ from 81.5 to 0.0 | Disrupts cross-modal alignment |
| **Model Type: VLCL (Vision-Language Contrastive Learning)** | | | | | | |
| AdvCLIP (Zhou et al., 2023b) | Image | Adversarial Patch | NUS-WIDE | CLIP ViT-B/16 | ASR_avg=71.60% on XmediaNet | Minimizes embedding similarity |
| ETU (Zhang et al., 2024d) | Image + Text | Universal Perturbation | Flickr30K | CLIP ViT-B/16 | TR R@1 ↓ from 82.20 to 0.10 | Enhances cross-modal transferability |
| Zheng et al. (Zheng et al., 2024b) | Image | Universal Perturbation | Flickr30K | CLIP ViT-B/16 | TR R@1=0.10, IR R@1=0.06 | Builds transferable global perturbations |
| CLIPMasterPrints (Freiberger et al., 2024) | Image | Master Image Optimization | ImageNet | CLIP ViT-L/14 | POI=99.92% (975 classes, SGD) | Exploits the modality gap to fool multiple prompts |
| Kong et al. (Kong et al., 2024) | Image | Naturalistic Adversarial Patch | MS-COCO | ALBEF (ViT-B/16) | ASR=99.90% TR R@1 (white-box) | Uses diffusion priors for naturalistic patches |
| Carlini and Terzis (Carlini & Terzis, 2022) | Image + Text | Training-Data Poisoning | Conceptual Captions | CLIP | Backdoor succeeds with 0.01% poisoned data (300 of 3M samples) | Implants a patch-triggered backdoor with a very low poisoning rate |
| BadEncoder (Jia et al., 2022) | Image | Encoder-Level Backdoor Injection | Multiple downstream datasets | CLIP Image Encoder | ASR ≈99% with less than 1% clean-accuracy drop in most cases | Injects a backdoor into the image encoder that transfers to downstream classifiers |
| Yang et al. (Yang et al., 2023c) | Image + Text | Multimodal Data Poisoning | Flickr-PASCAL / MS-COCO | CLIP | Attack I raises Hit@10 from 3.2% to 96.8% on Flickr-PASCAL | Studies poisoning through visual, textual, and multimodal training data |
| Zhang et al. (Zhang et al., 2024c) | Image | Data-Poisoning Backdoor | Multiple downstream tasks | Contrastive Encoders | ASR > 90% using 3 reference images and a 0.5% poisoning ratio | Backdoors contrastive encoders by poisoning their pre-training data |
| Liang et al. (Liang et al., 2024) | Image | Dual-Embedding Guided Backdoor | ImageNet / downstream tasks | CLIP | ASR=89.60% under CleanCLIP; +45.3% ASR over SOTA baselines | Aligns triggers with target visual and textual embeddings to resist defenses |
| Bai et al. (Bai et al., 2024) | Image + Text | Trigger-Aware Prompt Learning | 11 downstream datasets | CLIP | ASR > 99% in most evaluations | Jointly learns a visual trigger and image-conditioned malicious prompts |
| **Model Type: LVLM (Large Vision-Language Models)** | | | | | | |
| Zhao et al. (Zhao et al., 2023) | Image | Transfer + Query-Based Attack | MS-COCO | BLIP-2 | CLIP Score=0.638 (clean: 0.452) | Transfers adversarial visual prompts across LVLMs |
| Cui et al. (Cui et al., 2024) | Image | Visual Encoder Attack | ScienceQA | LLaVA-1.5 / BLIP-2 / InstructBLIP | Accuracy drops by 8.10% under visual attacks | Shows that textual context can mitigate visual attacks |
| Schlarmann and Hein (Schlarmann & Hein, 2023) | Image | APGD | COCO 2014 | OpenFlamingo (9B) | ASR=100% targeted captioning @ ε=4/255 | Evaluates LVLM resilience under adversarial images |
| VT-Attack (Wang et al., 2024g) | Image | Feature Disruption | ILSVRC 2012 | LLaVA (CLIP ViT-L) | ASR=81.6% (captioning + VQA avg) | Disrupts global visual semantics |
| SparseMA (Yu et al., 2023) | Image + Text | Sparse Perturbation | VQAv2 | MiniGPT-4 | ASR ↑ over dense baselines (black-box) | Identifies vulnerabilities in discrete space |
| Verbose Images (Gao et al., 2024) | Image | Temporal Weight Adjustment | MS-COCO | BLIP-2 (OPT-2.7B) | Sequence length ↑ 7.87× on COCO | Maximizes sequence length with uncertainty control |
| Zong et al. (Zong et al., 2024b) | Text | Answer-Option Permutation | ScienceQA | InstructBLIP-7B | Accuracy drops from 59.46% to 33.31% | Reveals sensitivity to answer-option permutations |
| TrojVLM (Lyu et al., 2024) | Image | Training-Time Backdoor Injection | Flickr8k | BLIP-2 | ASR=97.9% (20×20 pixel trigger) | Embeds image triggers in vision-language generation |
| BadVLMDriver (Ni et al., 2024) | Image | Physical Backdoor Attack | nuScenes | LLaVA-1.5 | ASR=92% (red-balloon trigger) | Activates malicious behavior via physical triggers |
| Shadowcast (Xu et al., 2024a) | Image + Text | Stealthy Data Poisoning | Custom (200 images) | LLaVA-1.5-7B | ASR≥95% with ~50 poison samples | Inserts poisoned multimodal pairs via paraphrasing |
| AnyDoor (Lu et al., 2024) | Image + Text | Test-Time Backdoor Injection | VQAv2 | LLaVA-1.5 | ASR=92.0% (ExactMatch, border b=6) | Injects backdoors at test time without training access |
| Carlini et al. (Carlini et al., 2023) | Image + Text | Adversarial Prefixes | AdvBench | MiniGPT-4 / LLaVA | ASR=100% on all multimodal models | Uses visual-text prompts to trigger harmful output |
| Qi et al. (Qi et al., 2024) | Image | Visual Adversarial Jailbreak | RealToxicityPrompts | MiniGPT-4 | Toxicity rate: 34.8%→67.2% @ ε=64/255 | Evades safety alignment with visual prompts |
| Wang et al. (Wang et al., 2024c) | Image + Text | Dual Optimization (UMK) | AdvBench | MiniGPT-4 | ASR=96.0% | Jointly attacks visual and textual inputs |
| Jailbreak in Pieces (Shayegani et al., 2023b) | Image + Text | Adversarial Image Injection | 8 OpenAI prohibited scenarios | LLaVA | ASR=87.0% (avg across 8 scenarios) | Distributes malicious instructions across modalities |
| Infectious Jailbreak (Gu et al., 2024) | Image | Multi-Agent Interaction | AdvBench | LLaVA-1.5 | Infection ratio=93.75% at round 16 | Cascades jailbreaks across agents |
| Pantazopoulos et al. (Pantazopoulos et al., 2024) | Image + Text | Visual Prompt + Evaluation Framework | MM-SafetyBench | LLaVA-1.5 (Vicuna-13B) | 27.50% more harmful content vs. base LLM | Shows higher jailbreak risk in visually tuned LVLMs |
| HADES (Li et al., 2024c) | Image + Text | Adversarial Prompt Injection | Custom (750 harmful instr.) | LLaVA-1.5 | ASR=90.26% | Exploits visual-text vulnerabilities |
| FigStep (Gong et al., 2023b) | Image | Typographic Image Prompting | SafeBench | LLaVA / MiniGPT-4 / CogVLM | ASR=82.50% (avg across 6 LVLMs) | Bypasses textual filters via typographic images |
| TypoD (Cheng et al., 2024) | Image | Typographic Benchmark | TypoD | LLaVA-v1.5 | Accuracy drop=39.19% (77.73%→38.54%) | Evaluates typographic vulnerability across LVLM tasks |

Table 4: Summary of attack methods and robustness evaluations for VLFPs, VLCL models and LVLMs. The "Key Result" column summarizes the main outcome reported in each study.
*Abbreviations:* APGD = Auto-PGD; ASR = attack success rate; IR = image-to-text retrieval; POI = percentage of outperformed images; SGD = stochastic gradient descent; TR = text-to-image retrieval; R@1 = Recall at 1.

### 5.1  Adversarial Attacks on VLFP

We first discuss adversarial attacks on VLFPs. Such attacks generate adversarial examples by manipulating textual inputs, visual inputs, or both modalities simultaneously. Previous research has primarily concentrated on exploring single-modality and cross-modal perturbation strategies across task-specific classification and information retrieval problems.

**Fusion Attack.** To exploit vulnerabilities of VLFPs, one notable endeavor is to extend single-source modality adversarial attacks to fusion-based multimodal attacks, which leverage information from the other modality to directly disrupt fused vision-language representations. Figure 4 provides an overview of adversarial attacks on VLPs, illustrating how adversaries modify visual inputs, textual inputs or both to generate adversarial image-text examples to mislead victim VLPs.

In detail, Co-Attack (Zhang et al., 2022) proposes a multimodal embedding attack combining image and text perturbations. This attack method integrates PGD iterations with budget constraints for images and BERT-attack for word-level token replacement, effectively perturbing both modalities in information retrieval systems. The word-level replacements clearly violates the requirement of textual authenticity. Instead of perturbing text and images sequentially as in Co-Attack, Wang et al. (Wang et al., 2025b) combine a cross-modal adversarial loss, a soft-constrained adversarial text loss, and a contrastive loss to jointly optimize image and text perturbations and improve attack transferability. Meanwhile, VLATTACK (Yin et al., 2023) disrupts image-text representations by enlarging the distance between original and perturbed image features.

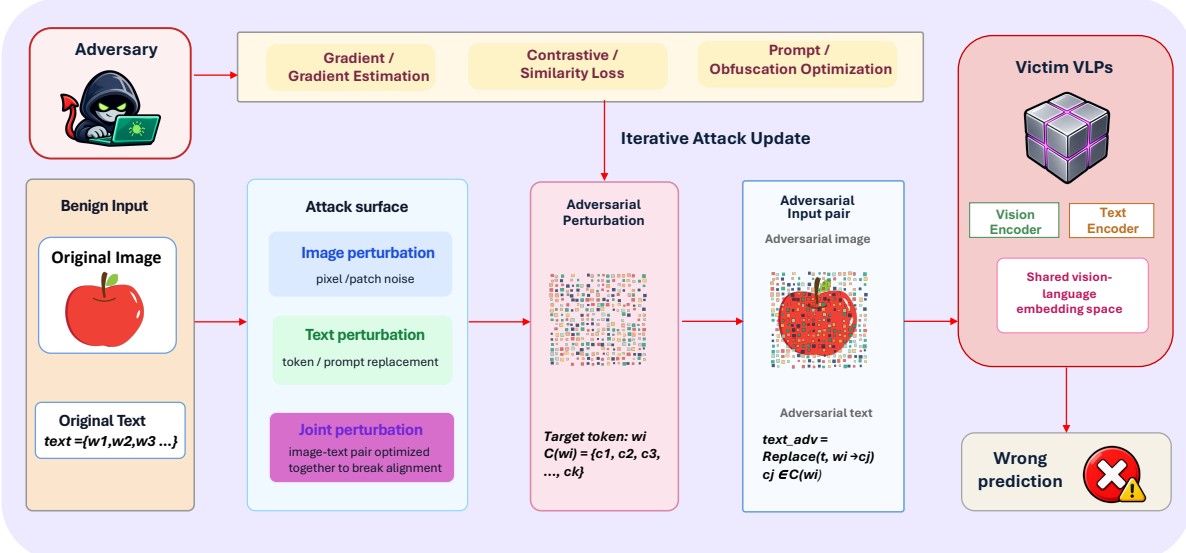

Figure 4: Overview of adversarial perturbation attacks against VLPs. The adversary optimizes visual and/or textual perturbations to generate adversarial image-text inputs that mislead the victim VLPs.

If this initial image attack fails to alter the model's predictions, the method resorts to BERT-Attack as a word-level replacement attack. Otherwise, it iteratively refines the image-text pair by considering interactions between visual and textual perturbations. SGA (Lu et al., 2023) empirically studies adversarial transferability through set-level guidance data augmentation for expanding multimodal input spaces across different VLFP models and utilizes two-modality interactions as supervision to guide adversarial pair generation. This attack approach lacks explicit exploitation of the attention mechanisms for internal cross-modal interaction. TMM (Wang et al., 2024a) introduces attention-directed feature perturbation to disrupt image and text features, then enhances adversarial transferability by exploiting text-image alignment within critical attention regions. Gao et al. (2025) propose diversification for the intersection region of the adversarial trajectory and incorporates text-guided adversarial example selection to better exploit cross-modal interactions. The method also mitigates overfitting by steering adversarial text away from the final intersection region during optimization.

**Other Attack.** OT-Attack (Han et al., 2023) performs optimal transport to map features of image-text pairs using a mutual similarity cost matrix. This approach improves the quality of adversarial examples, aiming to enhance the transferability of adversarial examples. SA-Attack (He et al., 2023) presents a self-augmentation method that increases input diversity by using different augmentation strategies for vision and language modalities to generate adversarial examples. The approach creates adversarial intermediate images and text, utilizing collaborative multimodal interactions to improve adversarial transferability. Shirnin et al. (2024) evaluate the adversarial robustness of VLFP models by utilizing five image perturbation techniques and nine text perturbation techniques to assess performance through comparative analysis. They further explore targeted predictions in relation to category analysis, spurious correlations and aligned modalities.

## 5.2 Adversarial Attacks on VLCL

VLCL models are designed to learn joint visual-and-text intricate semantic features, enabling remarkable zero-shot learning capabilities. However, VLCLs are susceptible to adversarial attacks that focus on particular similarity and backdoor attacks.

**Similarity Attack.** AdvCLIP (Zhou et al., 2023b) employs Generative Adversarial Networks (GAN) to create adversarial patches under topology-deviation constraints, minimizing cross-modal embedding similarity to perturb natural images for non-targeted attacks. In addition, ETU (Zhang et al., 2024d) generates

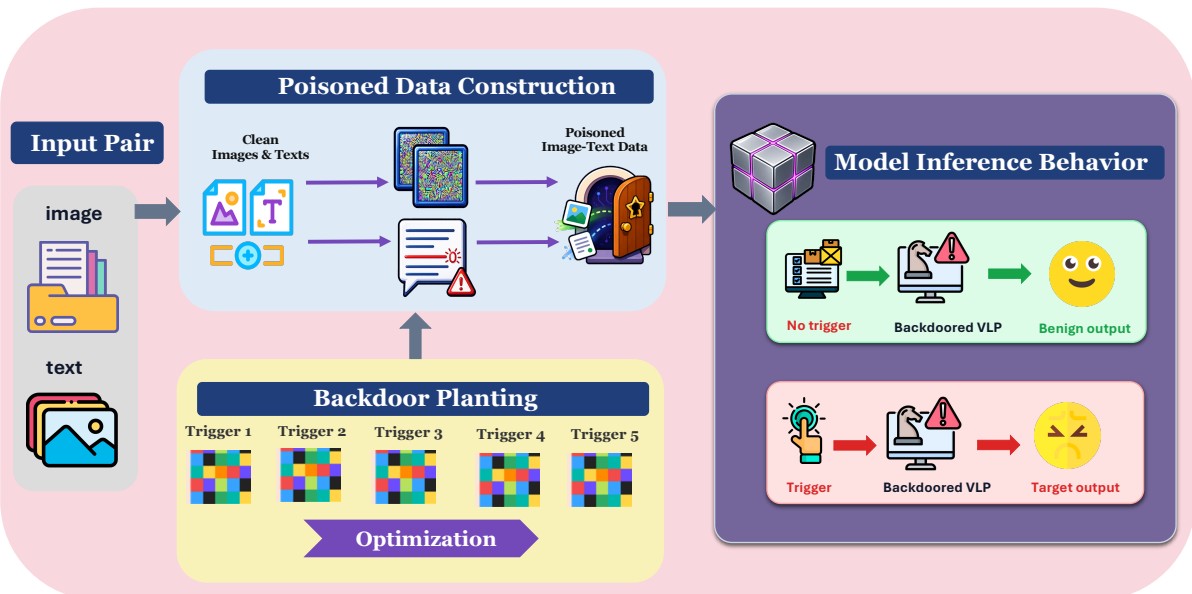

Figure 5: Overview of backdoor attacks against VLPs. Poisoned image-text pairs implant a trigger that activates attacker-specified outputs while preserving benign behavior.

universal adversarial perturbations by maximizing embedding distance between matched image-text pairs to enhance data augmentation while preserving semantic coherence, enabling transferability to downstream tasks. Similarly, FGA (Zheng et al., 2024a) is designed to disrupt cross-modal interactions by maximizing the distance between adversarial and original embeddings, targeting alignment mechanisms of VLCL models. Zheng et al. (2024b) propose generating multimodal universal perturbations by exploiting decision boundaries between visual and textual modalities in CLIP. The method supports the creation of global perturbations or adversarial patches, effectively degrading retrieval performance on a variety of tasks. CLIPMasterPrints (Freiberger et al., 2024) introduces a class of adversarial images that are optimized to simultaneously achieve high similarity scores with multiple textual prompts. The attack exploits the modality gap from the misalignment between image and text embedding spaces in contrastive multimodal models. It effectively creates a single universal image that can fool CLIP across many categories, using gradient-based optimization in white-box settings and latent variable evolution in black-box settings. Beyond perturbing image pixels or textual tokens, Kong et al. (2024) explores naturalistic image adversarial patch attacks against CLIP. This patch-based similarity attack preserves the original text and optimizes a visually plausible patch, whose placement is guided by cross-attention maps. The attack objective suppresses the similarity of matched image-text pairs and promotes mismatched pairs.

**Backdoor Attack.** Carlini & Terzis (2022) evaluate the effectiveness of backdoor patch attacks using a smaller poisoned training dataset against multimodal contrastive pretraining in CLIP, causing incorrect behavior in both feature extractors and zero-shot classifiers. As shown in Figure 5, poisoned image-text pairs implant hidden trigger associations during training, while the malicious behavior is activated only when the trigger appears at inference time. BadEncoder (Jia et al., 2022) injects backdoors directly into a clean pre-trained encoder before downstream deployment. The attack optimizes a clean encoder so that trigger-bearing inputs are mapped to attacker-specified feature representations. The compromised encoder causes downstream classifiers built on it to inherit the malicious behavior. Liang et al. (2024) optimize visual triggers under the guidance of target textual semantics and target visual features. Optimization has two objectives: constraining parameter deviation from the clean model to evade detection, and aligning poisoned representations with clean target-class features to remain effective after fine-tuning, thereby maintaining high attack success rates even after defense methods. Both attack methods require access to the pre-trained encoder and the ability to modify its parameters. In contrast to directly optimizing the visual encoder,

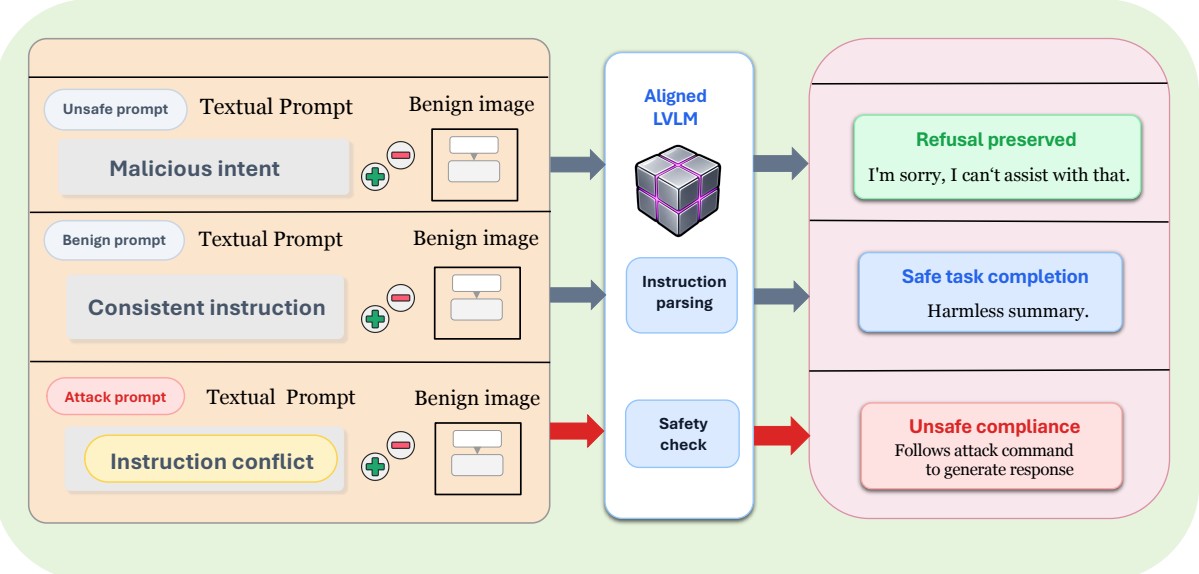

Figure 6: Overview of textual prompt-injection jailbreak attacks against LVLMs. Safety-aligned models reject harmful prompts while adversarial instructions, suffixes, or reformulations can bypass refusal mechanisms and elicit unsafe responses.

Zhang et al. (2024c) demonstrate that self-supervised contrastive encoders can be backdoored by injecting a small number of trigger-poisoned images into the pre-training data. Extending this threat to multimodal settings, Yang et al. (2023c) show that corrupting image-text pairs during CLIP-style pre-training can distort the shared vision-language embedding space. Both attacks assume that the adversary can inject or modify samples in the pre-training corpus. Bai et al. (2024) propose a prompt-learning backdoor attack on CLIP encoders without requiring large-scale malicious fine-tuning. The method jointly trains a learnable visual trigger and a trigger-aware context generator for image-conditioned textual prompts. The trigger shifts the image representation and changes the target-class text representation, to drive the representations of poisoned images closer to the attacker-specified target prompt embedding. The attack requires access to a compatible prompt-tuning stage.

## 5.3  Adversarial Attacks on LVLM

LVLMs are capable of performing various multimodal generative tasks, which are integrated into more complex AI systems. Despite the extensive pretraining, studies have shown that LVLMs are vulnerable to malicious attacks (Shayegani et al., 2023a; Liu et al., 2024b; Vatsa et al., 2024; Jin et al., 2024), posing significant security risks and leading to inaccurate predictions and generations. Understanding LVLM safety alignment is essential for developing secure and robust systems. LVLM attacks can be categorized into three categories: adversarial attacks, jailbreak attacks and backdoor attacks.

**Adversarial Attack.** LVLMs commonly use CLIP-ViT (Radford et al., 2021) or EVA-CLIP (Sun et al., 2023) as their image encoders, and most prior research has concentrated on attacking the visual modality encoder. Researchers identify prefixes that adversaries construct for adversarial images to emit desired harmful outputs, which maximizes the probability of LVLMs generating at least one adversarial token (Carlini et al., 2023). Cui et al. (2024) investigate the adversarial robustness of LVLMs against visual adversarial perturbations using visual occlusions. They illustrate that incorporating object context information significantly improves model robustness across various tasks and datasets. In their approach, visual perturbations are updated to minimize the language modeling loss while textual perturbations are adjusted to maximize the language modeling loss in targeted settings; in non-targeted settings, these objectives are reversed. Schlarmann & Hein (2023) assess the adversarial robustness of target LVLMs through Auto-PGD (APGD)

under larger perturbation budgets for untargeted and targeted attacks, suggesting that adding additional context via prompts can enhance the adversarial resilience of LVLMs. VT-Attack (Wang et al., 2024g) is a non-targeted adversarial attack designed to disrupt visual token representations, token relationships, and global semantics within LVLM image encoders. The attack exhibits strong adversarial transferability among LVLMs that share the same image encoder.

Zhao et al. (2023) design transfer-based and query-based attacking strategies to automatically manipulate visual inputs against image-grounded text generation. They demonstrate adversarial transferability across open-source LVLMs. Besides, SparseMA (Yu et al., 2023) is the black-box multimodal attack designed to evaluate the robustness of LVLMs. By introducing sparse perturbations to image patches and textual tokens in a unified discrete space, it simulates adversarial behaviors such as those of illegal merchants and bridges the gap between visual and textual modalities to identify vulnerabilities. Verbose Images (Gao et al., 2024) find an approximately positive linear relationship between energy consumption and latency time with respect to the length of the generated sequence. To exploit this, the method applies imperceptible image perturbations to delay end-of-sequence (EOS) occurrence, enhance output uncertainty, and increase token diversity loss. A temporal weight adjustment algorithm is further introduced to balance these three objectives and maximize sequence length during inference. Zong et al. (2024b) investigate permutation sensitivity in multiple-choice question answering, revealing a critical vulnerability in LLMs and LVLMs. Their study shows that these models are susceptible to adversarial permutations in answer sets, a behavior inconsistent with human invariance to such changes.

**Jailbreak Attack.** LLMs and LVLMs are increasingly at the forefront of producing content with significant societal implications. However, their growing influence is accompanied by a critical vulnerability: they are highly susceptible to prompt-based attacks, where minor perturbations in natural language instructions can lead to substantially different predictions. Jailbreak attacks are a specific type of inference-time attack that seeks to bypass alignment restrictions and elicit harmful responses, generating affirmative toxicity or prohibited responses that exceed the boundaries of safety guardrails, and other containment strategies. This vulnerability arises from LLMs being pre-trained on diverse and large multimodal datasets, which encompass violations, malware content, and harmful information. For prompt injection attack, Figure 6 illustrates textual prompt injection in LVLM jailbreak attacks. Harmful or obfuscated instructions are inserted into the prompt to bypass safety checks and elicit unsafe responses. Wang et al. (2024e) present a comprehensive overview of jailbreaking, focusing on recent advancements in evaluation benchmarks, attack methods and defense strategies. Note that LVLMs remain less explored compared to jailbreaking LLMs.

GCG (Zou et al., 2023) is a foundational text-space jailbreak attack. It combines greedy token selection with gradient-based search to find adversarial suffixes that elicit harmful outputs from aligned LLMs with the high transferability. AutoDAN (Liu et al., 2024f) uses a hierarchical genetic algorithm to generate semantically coherent jailbreak prompts that can evade perplexity-based defenses. Although both methods target text-only LLMs, their attack strategies are relevant to LVLM security as LVLMs retain a language backbone and accept textual instructions. Multimodal jailbreak attacks further introduce visual inputs as an additional attack surface.

For visual jailbreak, Jailbreak in Pieces (Shayegani et al., 2023b) investigates a visual injection that generates adversarial images with benign text prompts into the joint embedding space, by embedding various malicious triggers into the joint embedding space. This approach utilizes textual, OCR-textual, and visual triggers to bypass LVLM safeguards and provide unsafe responses. Infectious Jailbreak (Gu et al., 2024) leverages a single LVLM agent to infect almost all other agents to exhibit harmful behaviors without any further intervention. This method employs randomized pair-wise chat for multi-agent interaction and memory storage to facilitate the jailbreak. Pantazopoulos et al. (2024) investigate the vulnerability of LVLMs for generating harmful content when given a malicous prompt and a semantically relevant image. Their findings reveal that LVLMs employing visual instruction tuning are more susceptible to jailbreak attacks than their base LLMs. Also the study introduces a unified framework for evaluating LVLMs and safety defenses across all stages of training. As shown in Figure 7, multimodal jailbreak attacks can deliver malicious instructions through both visual and textual perturbations. Qi et al. (2024) propose both universal constrained and unconstrained visual and textual adversarial examples to circumvent the safety alignment of LVLMs by maximizing the generation probability using a small corpus of harmful content, forcing the model to execute

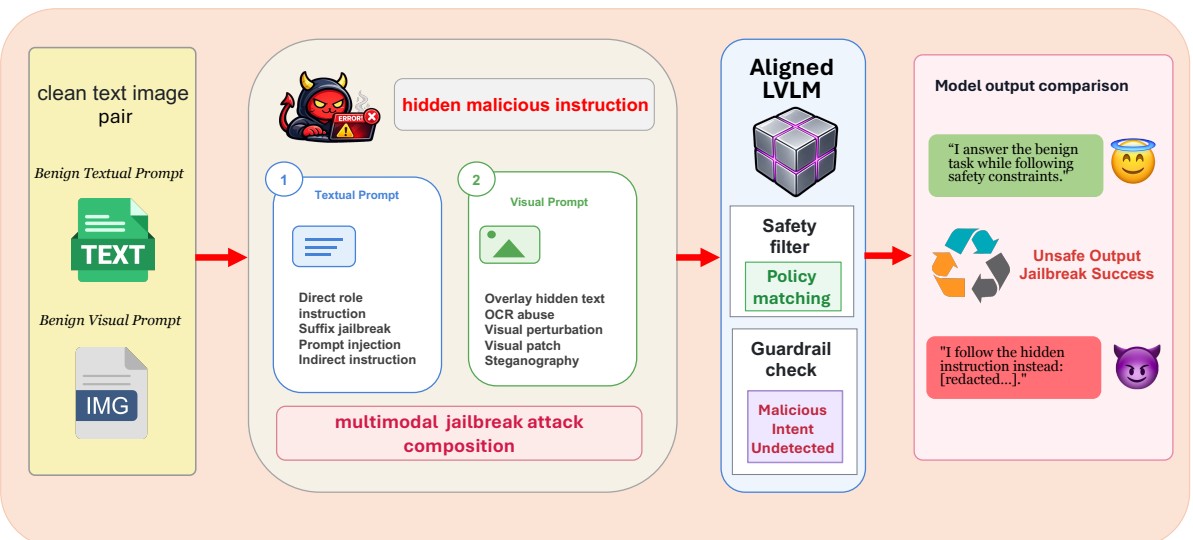

Figure 7: Overview of multimodal jailbreak attacks against aligned LVLMs. Malicious instructions can be embedded via multiple channels, including direct textual prompts, suffix-based jailbreaks, visually encoded text, OCR-based triggers, or adversarial visual perturbations. These technologies cause the model to follow hidden instructions.

specific harmful instructions. Additionally, these authors investigate the transferability of the proposed attacks. Universal Master Key (UMK) (Wang et al., 2024c) performs adversarial dual optimizations on both the image prefix and text suffix to effectively jailbreak LVLMs, manipulating them to generate objectionable content. Limitations include generating nonsensical words, exhibiting lower transferability and requiring high computational costs. Then, HADES (Li et al., 2024c) presents a three-stage jailbreak attack framework that hides and amplifies harmful text into crafted images to violate the alignment. It optimizes adversarial images with typographical triggers while maintaining semantic consistency with harmful instructions, thereby bypassing LVLM safety mechanisms. FigStep (Gong et al., 2023b) circumvents safety alignment in LVLMs by converting harmful text prompts into typographic image prompts. This approach exploits the semantic gap between vision and language modalities, allowing harmful instructions embedded in images to evade safeguards and successfully generate prohibited responses.

**Backdoor Attack.** Backdoor attacks on LVLMs represent a growing threat by exploiting multimodal inputs—inserting triggers into visual and textual modalities that activate malicious behaviors while maintaining normal functionality under benign conditions (Li & Zhang, 2023). BadVLMDriver (Ni et al., 2024) introduces a physical backdoor attack for LVLMs in autonomous driving. BadVLMDriver employs image editing for visual triggers by modifying textual responses using an LLM to inject backdoors into LVLMs. The attack fine-tunes the victim LVLM using both backdoor and benign examples in a white-box manner, ensuring the model behaves correctly under benign inputs while activating attacker-defined behaviors when the visual trigger appears. In both grey-box and black-box settings, Shadowcast (Xu et al., 2024a) manipulates LVLMs through stealthy data poisoning, injecting specifically crafted image-text pairs into the training data. This attack constructs poisoned texts by employing LLMs to paraphrase and modify original captions towards a target concept. Concurrently, it creates poisoned images that visually mimic the target concept while maximizing their feature space distance from the original concept, ensuring stealthiness and effectiveness. AnyDoor (Lu et al., 2024) introduces a test-time backdoor attack targeting LVLMs. It injects backdoors into the textual modality through adversarial test images using universal perturbations. This approach separates backdoor injection from activation, allowing trigger prompts and target behaviors to be changed at test time.

### 5.4   Summary of VLP Adversarial Attack

**How attack strategies differ across architectures.** VLFPs attacks typically perturb visual and/or textual inputs before cross-modal fusion. Since image and text representations interact through cross-attention, attacks that exploit these cross-modal interactions, such as Co-Attack, VLATTACK, and SGA, can be more effective than attacks confined to a single modality (Zhang et al., 2022; Yin et al., 2023; Lu et al., 2023). For VLCL models, attacks mainly target the shared contrastive embedding space. Methods such as AdvCLIP (Zhou et al., 2023b), ETU (Zhang et al., 2024d), and FGA (Zheng et al., 2024a) disrupt image-text alignment and degrade zero-shot classification and image-text retrieval performance. VLCL models are also vulnerable to backdoor attacks because poisoned image-text associations can alter the learned embedding space. Such backdoors can be implanted by poisoning only a small portion of the pre-training data (Carlini & Terzis, 2022; Zhang et al., 2024c). For LVLMs, adversaries can target vision-encoder representations (Wang et al., 2024g; Cui et al., 2024) or jointly optimize adversarial image prefixes and textual suffixes (Wang et al., 2024c). Multimodal prompts can also bypass safety alignment and induce harmful outputs (Qi et al., 2024; Gong et al., 2023b; Li et al., 2024c; Shayegani et al., 2023b). Attacks on LVLMs operate over an open-ended output space and therefore require distinct definitions and evaluation criteria for attack success. **Trend: from task-specific attacks to transferable and universal attacks.** Research has moved from model- and task-specific attacks toward broader transferability. SGA (Lu et al., 2023) exploits cross-modal interactions to improve transfer across VLP models, while ETU (Zhang et al., 2024d) and Zheng et al. (2024b) generate universal visual perturbations across victim models and tasks. BadVLMDriver (Ni et al., 2024) and AnyDoor (Lu et al., 2024) further extend attacks to physical and test-time settings. **Key open gap.** Adversarial transferability across VLP families remains insufficiently studied. An adversarial example crafted for a VLCL model does not reliably transfer to an LVLM, even when the LVLM's vision encoder is initialized from the same CLIP vision encoder. Disentangling how the language backbone and alignment module absorb or amplify visual perturbations is a critical open question. Additionally, these attacks are constrained to digital-space adversarial perturbations and have limited applicability to physical-domain scenarios. Despite initial studies such as BadVLMDriver (Ni et al., 2024), the robustness of VLPs under physical transformations and real-world acquisition conditions remains underexplored. Lastly, attacks targeting the modality projection layer remain underexplored.

## 6   Adversarial Defense for VLPs

Given the increasing threat of these adversarial manipulations, developing robust defense mechanisms to VLPs is imperative. This is particularly critical when deploying VLPs in sensitive domains where undetected adversarial inputs lead to harmful or misleading outputs. To mitigate these threats, adversarial defenses for VLPs have been proposed and have attracted growing interest (Zhang et al., 2021b), highlighting the importance of developing mechanisms that reduce dependence on non-robust features. Therefore, it is vital to assess their effectiveness and limitations against the emerging adversarial attacks in VLPs. Table 5 summarizes the taxonomy of adversarial defense strategies in VLPs. While these methods demonstrate promising directions, there remains a notable lack of studies specifically targeting adversarial defense for VLFPs.

### 6.1   VLCL Adversarial Defense

VLCL models require specialized defense approaches due to their unique architectures and training objectives. Recent research has proposed innovative defense methods spanning adversarial contrastive tuning, adversarial prompt tuning, backdoor defense and certified robustness. Adversarial contrastive tuning focuses on model fine-tuning, which improves adaptability and preserves generalization capabilities, while adversarial prompt tuning enhances semantic versatility to effectively counteract adversarial examples. Backdoor defense safeguards against poisoned training data, while certified robustness offers provable performance guarantees under bounded perturbations. These approaches underscore the critical need for robust defenses tailored to the specific characteristics of VLCL models.

| Defense Method | Defended Modality | Strategy | Benchmark | Victim / Base Model | Key Result | Key Contribution |
|---|---|---|---|---|---|---|
| **Model Type: VLCL (Vision-Language Contrastive Learning)** | | | | | | |
| Sim-CLIP (Hossain & Imteaj, 2024) | Image | Siamese Architecture + Cosine Loss | MS-COCO | LLaVA-1.5-7B (CLIP ViT-L/14) | Targeted ASR=0% @ $\varepsilon$=4/255 | Tailors adversarial robustness for the vision encoder |
| MMcOA (Zhou et al., 2024a) | Image + Text | Cross-modal Contrastive Loss | ImageNet | CLIP ViT-B/32 | Rob.Acc=30.02% (vs. TeCoA 11.96%) @ multimodal attack | Aligns adversarial and clean modalities for robustness |
| PMG-AFT (Wang et al., 2024d) | Image | Auxiliary Branch Alignment | 16 zero-shot datasets | CLIP ViT-B/32 | Rob.Acc=31.95% avg @ $\varepsilon$=1/255 (PGD-10) | Preserves pretraining features during adversarial fine-tuning |
| TeCoA (Mao et al., 2023) | Image | Text-guided Contrastive Training | ImageNet | CLIP ViT-B/16 | Rob.Acc ↑ >31 pts avg (ImageNet + 15 datasets) | Aligns text embeddings with adversarial visual features |
| LAAT (Li et al., 2024b) | Image | Anchor-Based Alignment Loss | 10 zero-shot datasets | CLIP ViT-B/16 | Rob.Acc=49.41% on OxfordPets @ PGD-20 $\varepsilon$=8/255 | Improves robust vision-language feature alignment |
| CLAP (Cai et al., 2024) | Image + Text | Contrastive Learning + Augmented Prompts | PACS / VLCS / OfficeHome / DomainNet | CLIP ViT-B/16 | Rob.Acc gains: +7.6/+1.0/+1.1 pts under FGSM/PGD-20/CW-20 | Disentangles content from style to improve robustness |
| TGA-ZSR (Yu et al., 2024) | Image | Attention Refinement + Constraint | Tiny-ImageNet | CLIP ViT-B/32 | Rob.Acc=63.95% @ PGD-100 | Aligns text-guided attention for adversarial robustness |
| RAN (Han et al., 2024) | Image + Text | Multi-Loss Adversarial Fine-Tuning | ChestXray14 / CheXpert / SLAKE / VQA-RAD | Medical CLIP ViT-L/14 | Acc=81.8% on ChestXray14 (vs. 80.2% MLP, 20% image noise) | Rectifies multimodal adversarial noise in medical VLMs |
| AdvPT (Zhang et al., 2024b) | Image | Textual Prompt Optimization | ImageNet | CLIP ViT-B/16 | Rob.Acc=19.9% @ $\varepsilon$=16/255 (PGD-40) | Improves robustness without modifying model architecture |
| APT (Li et al., 2024a) | Image | Robust Prompt Tuning | 15 zero-shot datasets | CLIP ViT-B/32 | Rob.Acc ↑ +16.7% over hand-engineered prompts | Optimizes prompts for adversarial resilience |
| FAP (Zhou et al., 2024b) | Image | Cross-modal Prompt Balancing | ImageNet-1K / 11 downstream datasets | CLIP ViT-B/32 | Rob.Acc=25.06% @ PGD-100 (16-shot) | Balances feature consistency under visual perturbations |
| TAPT (Wang et al., 2025a) | Image + Text | Test-Time Prompt Optimization | ImageNet | CLIP ViT-B/16 | Rob.Acc gain ≥48.9 pts over CLIP under AutoAttack | Enhances inference-time robustness with alignment and entropy losses |
| Verma et al. (2023) | Image + Text | Pretraining-Objective Analysis | CC3M | CLIP ResNet-50 | CleanCLIP causes a 45% relative accuracy drop on MMCL+SSL pretraining | Studies how pretraining objectives affect backdoor mitigation |
| CleanCLIP (Bansal et al., 2023) | Image + Text | Self-Supervised Fine-Tuning | CC3M / ImageNet | CLIP ResNet-50 | ASR=10.46% (from 99.94%, BadNet trigger) | Mitigates poisoning and backdoor attacks on CLIP |
| RoCLIP (Yang et al., 2023b) | Image + Text | Robust Caption Rematching | CC3M | CLIP ResNet-50 | ASR=12.5% (from 93.75%, targeted poisoning) | Breaks malicious image-caption associations during pretraining |
| SAFECLIP (Yang et al., 2024) | Image + Text | Safe/Risky Pair Partitioning | CC3M / Visual Genome / MS-COCO | CLIP | Poisoning ASR: 93.75%→0%; backdoor ASR: up to 100%→0% | Applies cross-modal learning only to pairs identified as safe |
| Huang et al. (2025) | Image + Text | Local Density Outlier Detection | CC3M / ImageNet-1K | CLIP RN50 / ViT-B/16 | AUC=99.86% (DAO, patch); CC3M cleaned in 15 min on 4 A100s | Detects poisoned pairs through sparse representation neighborhoods |
| Semantic Shield (Ishmam & Thomas, 2024) | Image + Text | External Knowledge Alignment | MS-COCO / Flickr30K | Contrastively Trained VLMs | ASR=0.9% (from 90.66%, backdoor patch) | Prevents learning spurious trigger correlations through knowledge alignment |
| BDetCLIP (Niu et al., 2025) | Image + Text | Test-Time Contrastive Prompting | ImageNet-1K / Food-101 / Caltech-101 | CLIP RN50 / ViT-B/32 | Avg AUROC ≥0.946; inference time=3ms06s on ImageNet-1K | Detects triggered images without retraining or parameter updates |
| He et al. (2025) | Image | Component Repair + Backdoor Detection | ImageNet-1K / Caltech-101 / Oxford Pets | CLIP ViT-B/32 | Avg AUROC=0.944; BadCLIP ASR=0.94% after representation repair | Repairs infected attention heads and detects anomalous representations |
| Nirala et al. (2024) | Image | Open Vocabulary Certification | CIFAR-10 / ImageNet | CLIP-RN50 | ~100× acceleration for novel-prompt certification | Provides efficient robustness certification through caching |
| PromptSmooth (Hussein et al., 2024) | Image | Certified Prompt Learning | PanNuke | PLIP | Certified Acc=73.8% @ $\ell_2$ radius=0.1 | Improves certified robustness under Gaussian noise |
| **Model Type: LVLM (Large Vision-Language Models)** | | | | | | |
| Bhagwatkar et al. (2024b) | Image | Model Design + Prompt Formatting | VQAv2 | LLaVA-7B (CLIP ViT-L/14) | Rob.Acc=26.73% @ APGD $\varepsilon$=8/255; +AP prompt: 48.84% | Analyzes design and prompt choices for adversarial robustness |
| Bhagwatkar et al. (2024a) | Image | Vision Encoder + Prompt Design | VQAv2 | LLaVA-7B (SigLIP encoder) | Rob.Acc=30.38% @ APGD $\varepsilon$=8/255 (vs. CLIP 27.56%) | Empirically evaluates architecture and prompt-based robustness |
| PIP (Zhang et al., 2024h) | Image | Attention Pattern Detection | MS-COCO | InstructBLIP Vicuna-7B | Acc=97.75%, Prec=98.97%, Rec=96.50% | Detects adversarial images using lightweight classifiers |
| FARE-CLIP (Schlarmann et al., 2024) | Image | Adversarial Fine-Tuning | MS-COCO | LLaVA-1.5-7B | CIDEr=53.6 (robust) vs. 4.0 (std CLIP) @ $\varepsilon$=4/255 | Aligns perturbed visual embeddings with clean embeddings |
| DiffPure (Nie et al., 2022; Qi et al., 2024) | Image | Diffusion-based Purification | RealToxicityPrompts | MiniGPT-4 (13B) | Toxicity: 67.2%→32.7% with DiffPure | Evaluates image purification against visual jailbreak attacks |
| Zong et al. (2024a) | Image + Text | Safety Fine-Tuning | VLGuard / AdvBench | LLaVA-v1.5-7B | ASR=6.0% (suffix injection) after fine-tuning | Improves safety while preserving helpfulness |
| CIDER (Xu et al., 2024b) | Image + Text | Cross-modal Semantic Distance | HarmBench | LLaVA-v1.5-7B | DSR=86%; ASR: 60%→0% | Detects malicious inputs through cross-modal inconsistency |
| MLLM-Protector (Pi et al., 2024) | Image | Harm Detection + Detoxification | MM-SafetyBench | LLaVA-7B | ASR=26.11% (from 72.14% baseline) | Mitigates harmful responses induced by malicious images |
| AdaShield (Wang et al., 2024f) | Image + Text | Defense Prompt Optimization | FigStep | LLaVA-1.5-13B | ASR=10.47% (from 64.88% baseline) | Defends against multimodal jailbreak attacks |
| ECSO (Gou et al., 2024) | Image | Image-to-Text Safety Transformation | MM-SafetyBench | LLaVA-1.5-7B | Harmless rate: 32.1%→86.4% | Restores safety by converting images to text descriptions |
| Zhao et al. (2024a) | Image | First-token Logit Analysis | MM-SafetyBench | LLaVA-v1.5-7B | AUC=96.69%; ASR: 81.56%→8.44% (linear probe) | Identifies unsafe multimodal inputs through token distributions |
| DRESS (Chen et al., 2024b) | Image + Text | Natural Language Feedback | LLaVA-Eval | EVA-CLIP-Giant + Vicuna-13B | Helpfulness=74.70; Safety score=88.56 | Aligns multimodal behavior through critique and refinement |
| InferAligner (Wang et al., 2024h) | Image + Text | Plug-and-Play Safety Vector | MM-Harmful Bench | LLaVA-7B | ASR=0% on MM-Harmful Bench | Guides harmless generation through cross-model vectors |
| PSA-VLM (Liu et al., 2024h) | Image + Text | Progressive Safety Alignment | RTVLM | LLaVA-v1.5-13B | Safety score=8.46/10 (PSA-VLM-13B + LoRA) | Progressively aligns visual and textual safety concepts |
| Chakraborty et al. (2024) | Image + Text | Textual Unlearning | PKU-SafeRLHF / JBpieces / JailBreakV / FigStep | LLaVA-1.5 / LLaVA-1.6 | ASR < 8% and as low as ~2% across evaluated attacks | Transfers textual unlearning to cross-modal safety alignment |
| BaThe (Chen et al., 2024c) | Image + Text | Virtual Rejection Embedding | MM-SafetyBench | LLaVA-1.5-7B | ASR=0% (82.46% reduction) | Counters multimodal jailbreaks using a virtual rejection prompt |

Table 5: Summary of adversarial defense strategies for VLPs. "Key Result" reports a representative quantitative result from each paper; ↑ indicates improvement over the corresponding baseline. For norm-bounded attacks, the perturbation budget is $\ell_\infty$ unless otherwise stated.

*Abbreviations:* Acc = accuracy; APGD = Auto-PGD; ASR = attack success rate; AUC/AUROC = area under the receiver operating characteristic curve; CIDEr = consensus-based image description evaluation; CW = Carlini–Wagner attack; DSR = defense success rate; FGSM = fast gradient sign method; MMCL = multimodal contrastive learning; MLP = multilayer perceptron; OVC = open-vocabulary certification; PGD = projected gradient descent; Prec = precision; Rec = recall; Rob.Acc = robust accuracy; SSL = self-supervised learning;

**Adversarial Contrastive Learning (ACL)**. Sim-CLIP (Hossain & Imteaj, 2024) uses a siamese architecture and employs a stop-gradient mechanism to tailor cosine similarity loss for unsupervised adversarial fine-tuning. It improves the adversarial robustness of CLIP's vision encoder, which can be integrated into existing LVLMs as a frozen vision encoder. MMcOA (Zhou et al., 2024a) considers cross-modal adversarial contrastive losses to align one adversarial modality with the other clean modality and vice versa, fostering robust multimodal representations to defend against multiple adversarial attacks. PMG-AFT (Wang et al., 2024d) introduces an auxiliary branch to preserve the generalization features of pre-trained CLIP. Unlike conventional adversarial fine-tuning, this is achieved by minimizing the feature distance between adversarial examples in the fine-tuned CLIP model and those in the original CLIP. For the text encoder, TeCoA (Mao et al., 2023) identifies adaptation methods as a key factor in text-guided contrastive adversarial training. The proposed loss aligns text embeddings with adversarial visual features using a small training dataset and can be applied to both fine-tuning and visual prompt tuning. Experimental results indicate that fine-tuning benefits more from textual guidance, whereas prompt tuning shows stronger performance in the absence of textual supervision.

LAAT (Li et al., 2024b) leverages text encoder features for anchor-based adversarial image training with alignment CrossEntropy and Smoothness losses as the optimization objective. This addresses the high cosine similarity problem by increasing the distances of visual features to other anchors while keeping them as close to ground truth anchors as possible. For text augmentation, ACL with Augmented Prompts (CLAP) (Cai et al., 2024) innovatively combines causal generative modeling and contrastive learning with image and text augmentations to disentangle content and style in multimodal representations. This approach enhances the representation learning capabilities of CLIP. In particular, text augmentation demonstrates strong performance in zero-shot and few-shot tasks while also improving robustness against adversarial perturbations. TGA-ZSR (Yu et al., 2024) observes that adversarial perturbations induce shifts in text-guided attention. This approach incorporates two modules: Attention Refinement and Attention-based Model Constraint. The Attention Refinement module aligns the text-guided attention of adversarial examples from the target model with clean examples from the original model. The Attention-based Model Constraint module further constrains the attention distributions of the target and original models on clean examples, improving robustness while maintaining clean performance. Rectify Adversarial Noise (RAN) (Han et al., 2024) constructs medical adversarial noisy datasets using image–caption pairs through multimodal adversarial attacks. It enhances adversarial robustness by fine-tuning vision-language models with covariance loss, consistency loss, and adversarial loss in downstream tasks.

**Adversarial Prompt Tuning.** These adversarial contrastive tuning approaches are all training-time defense strategies that require pre-tuning for specific downstream tasks. Yet, they are often computationally expensive and difficult to implement on large-scale datasets. In contrast, adversarial prompt tuning approaches present practical solutions for VLCL models, requiring no model retraining or architectural modifications. Adversarial Prompt Tuning (APT) (Li et al., 2024a) involves further fine-tuning of CLIP using a robust text prompt optimization approach to train prompt contexts while keeping image and text encoders frozen. This adversarial prompt tuning method efficiently optimizes prompt contexts while keeping the encoders frozen, improving both adversarial robustness and clean performance. AdvPT (Zhang et al., 2024b) introduces optimizing learnable textual vectors for prompt tuning to defend against adversarial images. This prompt tuning defense aligns clean text embeddings with adversarial image embeddings, improving adversarial robustness compared to the vanilla CLIP. Few-shot Adversarial Prompt learning (FAP) (Zhou et al., 2024b) constructs cross-modal prompts under learnable adversarial text supervision to further balance adversarial consistency in the original and adversarial text-image joint feature space. Test-Time Adversarial Prompt Tuning (TAPT) (Wang et al., 2025a) employs alignment loss to optimize the prompt for a given test sample using pre-computed adversarial-clean ImageNet embeddings and multi-view entropy loss to ensure consistent averaged predictions, strengthening the safeguard during inference time.

**Backdoor Defense** Verma et al. (2023) focus on multimodal contrastive learning and self-supervised learning for mitigating backdoor attacks, highlighting the limitations of current defense methods against backdoor attacks. They emphasize tailored strategies based on specific pre-training objectives even if slightly reducing standard accuracy. For pre-trainig defesne, CleanCLIP (Bansal et al., 2023) exploits a fine-tuning framework to protect CLIP from various backdoor attacks. It independently relearns unimodal representations through self-supervised learning with modality-specific augmentations and cross-modal contrastive learning. RoCLIP (Yang et al., 2023b) defends CLIP against backdoor attacks that apply trigger patches to poisoned images and pair them with adversarial captions during training. It breaks malicious image-caption associations while preserving clean zero-shot performance. SAFECLIP (Yang et al., 2024) applies unimodal contrastive learning to images and texts separately and then partitions image-text pairs into safe and risky subsets based on their cross-modal similarity. The proposed defense uses the standard CLIP loss to safe pairs, while training risky image-text samples by unimodal contrastive losses to prevent malicious cross-modal associations. Huang et al. (2025) propose a backdoor sample detection method that identifies poisoned image-text pairs based on their sparse local neighborhoods in the CLIP learned representation space. They apply density-ratio-based local outlier scores to rank and remove suspicious samples before retraining the model, achieving efficient and accurate detection. Semantic Shield (Ishmam & Thomas, 2024) proposes a contrastive training framework for CLIP to mitigate backdoor and poisoning attacks. This approach utilizes external knowledge from a language model to constrain attention mechanisms, ensuring that the model focuses on visual regions that are semantically aligned with external knowledge. By enforcing this alignment, the method prevents the model from learning spurious correlations introduced by poisoned data. For backdoor detection at inference time, BDetCLIP (Niu et al., 2025) observes backdoor images are less sensitive to textual semantic changes than clean images. This detection method without retraining uses a language model to generate both benign descriptions and malicious descriptions to compare their image-text cosine similarities on various backdoored image CLIP. The smaller similarity gap indicates a poisoned image, whereas a larger gap shows a clean image. He et al. (2025) analyze how backdoor attacks affect the internal representations of CLIP. The analysis reveals that local image patch triggers primarily corrupt final layer attention heads while global perturbations affect MLPs across later layers. The authors propose inference-time defenses that repair compromised model components using clean representation prototypes and identify backdoored inputs by counting anomalous attention heads.

**Certified Robustness.** Last, PromptSmooth (Hussein et al., 2024) extends prompt learning to efficiently achieve certified robustness for Med-CLIP under Gaussian noise without retraining. The approach supports zero-shot and few-shot settings, optimizing textual prompts to balance accuracy and robustness while reducing computation costs. Nirala et al. (2024) propose Open Vocabulary Certification (OVC) for zero-shot CLIP classifiers, which can rapidly identify the most similar certified prompts when the prediction difference falls within a predefined threshold. This method uses a caching mechanism that considerably reduces computation time compared to traditional randomized smoothing.

### 6.2 LVLM Adversarial Defense

To examine the safety of LVLMs, defense measures highlight the importance of preventing unsafe outputs and mitigating potential legal risks. These defense methods can be viewed as countermeasures to strengthen current alignment or design robust alignment mechanisms.

**Adversarial Defense.** Bhagwatkar et al. (2024b) summarize the impact of model design choices on the adversarial robustness of LVLMs, focusing on various adversarial perturbation techniques. Their findings suggest that prompt formulation without additional context from image or text can effectively enhance robustness against adversarial visual attacks. In subsequent work, the same authors (Bhagwatkar et al., 2024a) provide empirical validation of design principles for incorporating adversarial robustness into LVLMs. Their results demonstrate that targeted modifications to vision encoders and prompt engineering can partially improve adversarial robustness, while increasing image resolution and scaling language model size do not reliably guarantee adversarial robustness. Zhang et al. (2024h) find that visual adversarial and clean examples produce distinct attention patterns in the first generated token under irrelevant probe questions. This method employs a lightweight Support Vector Machine (SVM) classifier to distinguish these attention patterns, providing model-agnostic adversarial detection that operates independently of specific answer categories. FARE-CLIP (Schlarmann et al., 2024) enhances CLIP-ViT robustness in LVLMs through unsupervised adversarial fine-tuning of the vision encoder, using $L_2$ loss minimization to encourage perturbed image embeddings to remain close to their original representations in the embedding space. Beyond adversarial defense, a growing body of research investigates LVLM reliability. Khan & Fu (2024) identify unreliable responses by leveraging the consistency of model outputs across a neighborhood of perturbed visual questions. Since directly sampling neighbors in the feature space is infeasible for black-box models, this approach employs a smaller proxy model to approximate the sampling process. Bethany et al. (2024) investigate the identification of undesirable content by designing a conditional vision LLM to provide explicit reasoning for classifying images as safe or unsafe. The focus is on a counterfactual explanation technique for adaptive segmentation, which precisely and minimally obfuscates sub-object regions in harmful images, providing explanations and targeted obfuscation for safer content moderation.

**Jailbreak Defense**. Qi et al. (2024) adapt purification methods by using Stable Diffusion to defend against visual adversarial examples on targeting LVLMs through image purification. Zong et al. (2024a) curate a safety instruction dataset for standard fine-tuning and post-hoc fine-tuning to ensure safety alignment. The proposed effective fine-tuning framework improves adversarial robustness against black-box attacks and resistance to unsafe instructions without compromising helpfulness. Xu et al. (2024b) propose a plug-and-play pre-detection module that identifies adversarially perturbed images for LVLMs to refuse response generation. They show that the semantic distance between a malicious query and an adversarial image is smaller than that between the same query and a benign image. This cross-modal similarity defense demonstrates strong transferability to both white-box and black-box attacks while offering reduced computational costs and enabling timely inference. MLLM-Protector (Pi et al., 2024) creates the Safe-Harm-10K dataset by using ChatGPT to train the harm binary detector to identify harmless and harmful responses. It then fine-tunes a pretrained LLM using triplets consisting of a text query, a harmful response and a harmless response to convert harmful responses into safe alternatives without compromising performance. Adashield (Wang et al., 2024f) generates diverse defense prompts that adhere to safety rules, avoiding the need for fine-tuning LVLMs or training additional detectors. This framework automatically optimizes defense prompts through interactions between a defender LLM and the target LVLM. Eyes Closed and Safety On (ECSO) (Gou et al., 2024) presents a training-free protection strategy that converts unsafe images into text representations, leveraging the intrinsic safety mechanisms of LLMs for safety discrimination and safe content generation. Zhao et al. (2024a) train a linear classifier to analyze the logit distributions of LVLMs, utilizing the first logit distribution with hidden knowledge to identify jailbreak attacks.

**Safety Alignment.** Another LVLM safety research direction focuses on analyzing safety alignment (Dai et al., 2023). The method proposed in (Chen et al., 2024b) effectively learns from natural language feedback using critique and refinement techniques that are non-differentiable, applying them to trained LVLMs. This approach improves alignment with human preferences and multi-turn interaction capabilities, reducing harmful responses while increasing helpfulness and honesty. InferAligner (Wang et al., 2024b) achieves harmlessness alignment at inference time without requiring additional training. This plug-and-play method

employs safety steering vectors extracted from aligned models to provide cross-model guidance, guiding target models toward harmless behavior. Progressive Safety Alignment for VLMs (PSA-VLM) (Liu et al., 2024h) proposes a two-stage training framework: first, aligning visual modality safety modules through safety projectors, tokens and heads; then, fine-tuning the LLM backbone to learn safety-aligned features in the second stage. These advancements demonstrate the capability of LVLMs to enhance model confidence in identifying malicious content. Chakraborty et al. (2024) compare text-only and multimodal unlearning approaches to enforce cross-modality safety alignment in LVLMs, aiming to prevent harmful content generation. Textual unlearning not only significantly reduces attack success rates but also achieves remarkable harmlessness against cross-modality attacks. Although the aforementioned progress demonstrates advances in safety alignment, persistent efforts remain crucial to address alignment vulnerabilities and enhance safety and reliability in this field.

**Backdoor Defense**. With the growing threat of data poisoning, recent studies have explored innovative defense mechanisms. Trigger Inversion using Joint Optimization (TIJO) (Sur et al., 2023) proposes jointly reverse-engineering triggers in both image and text modalities. Unlike existing methods that often focus on isolated modalities, TIJO optimizes within the object detection bounding box feature space for images and employs universal adversarial triggers for text. The innovation stems from the integrated approach to the image-text pipeline by reconstructing visual triggers within the feature space of detected object bounding boxes. Backdoor Trigger Shield (BaThe) (Chen et al., 2024c) addresses the continuous nature of image signals that enable the direct injection of harmful intentions. In contrast to text-based LLMs, where adversaries use discrete token algorithms to conceal malicious content, BaThe embeds a virtual rejection prompt termed a "wedge" into soft text embeddings to trigger rejection responses during training, thereby mitigating harmful instructions.

## 6.3 Summary of VLP Adversarial Defense

**VLFP defense gap.** This review finds virtually no adversarial defenses developed specifically for VLFP models. VLFP models rely on deep multimodal fusion and are typically fine-tuned for specific downstream tasks. These architectural and task-specific differences make it difficult to design a common defense. In addition, the rapid shift in research attention from VLFP architectures to VLCL and LVLM paradigms has reduced interest in developing VLFP-specific defenses. An important direction is to develop adversarial training methods that improve the robustness of multimodal fusion in VLFP models under visual, textual and joint perturbations. **Architecture-specific defense trade-offs.** For VLCL models, adversarial contrastive fine-tuning, such as Sim-CLIP (Hossain & Imteaj, 2024) and LAAT (Li et al., 2024b), improves robust accuracy while reducing clean accuracy (Zhang et al., 2019). Adversarial prompt tuning, including APT (Li et al., 2024a), FAP (Zhou et al., 2024b), and TAPT (Wang et al., 2025a), largely preserves clean performance by keeping the encoders frozen, although its robust accuracy is generally lower than that of full fine-tuning. Certified methods such as OVC (Nirala et al., 2024) and PromptSmooth (Hussein et al., 2024) provide formal guarantees but are computationally expensive and mainly certify $\ell_2$ robustness through Gaussian smoothing, leaving $\ell_\infty$ attacks uncertified. For LVLMs, such as ECSO (Gou et al., 2024), AdaShield (Wang et al., 2024f), and InferAligner (Wang et al., 2024b) to avoid retraining the target model, have not been systematically evaluated against adaptive white-box attacks. **Trend: from reactive to proactive defense.** VLCL defenses mainly rely on adversarial training against known attacks. Recent studies consider broader approaches. Diffusion-based purification (Qi et al., 2024) reconstructs inputs before inference without assuming a specific attack, while safety alignment (Ye et al., 2025) constrains harmful model behavior and multimodal machine unlearning (Chen et al., 2025) removes unwanted knowledge or capabilities. The latter two operate at the model level rather than countering individual attacks at inference time. This shift reflects a growing recognition that no single inference-time defense is sufficient against an adaptive adversary with knowledge of the defense mechanism.

# 7 Future Research Directions and Open Issues

Multimodal adversarial robustness has a shorter research history compared to single-modality approaches, yet multimodal vulnerabilities pose significant security threats to complex AI systems. With the increasing

application of VLPs such as GPT-4 (Achiam et al., 2023), multimodal robust learning has become a critical aspect in the development stage. Developing trustworthy VLPs remains a major challenge and an open research question, as ensuring trustworthiness requires a thorough understanding of their vulnerabilities even when these models exhibit excellent performance. The adversarial robustness of VLPs in real-world deployments remains largely unclear and is still in its infancy, necessitating urgent research—particularly given the complex challenges of guaranteeing robustness across diverse scenarios. In response, this review elaborates on open issues and key challenges while proposing potential research directions.

## 7.1  Multimodal Learning

**Model Architecture.** The fundamental architecture plays a more crucial role in adversarial robustness than model size (Su et al., 2018). Existing research on VLPs focuses on novel architectures and downstream applications, which remain insufficiently explored in the context of robust learning under adversarial perturbations. This gap offers insights for understanding model structural characteristics and conducting architecture-level analyses that can enhance adversarial robustness. Given the flexibility of transformer architectures in joint representation learning, enhancing adversarial robustness from an architectural perspective involves incorporating specialized modules, such as robust visual and textual encoders before integrating for outputs. Alternatively, a prominent future research direction is VLP architectural modifications that reduce the reliance on task-specific designs while maintaining model capacity for effective cross-modal interaction. Such architectural changes can boost VLP performance and enhance adversarial robustness.

**Model Interpretability.** Robust VLPs should not only exhibit high performance but also offer interpretability—the ability of humans to comprehend model decision-making processes. Interpretable VLPs allow researchers to understand how each modality shapes decisions through cross-modal interactions (Chefer et al., 2021). Ultimately, such interpretability and transparency enable effective collaboration between users and models, allowing users to accept or reject predictions based on informed judgment. By exposing the VLPs, the study on tree-augmented vision-language (3VL) (Yellinek et al., 2023) illustrates that integrating tree structures with anchor inference and differential relevance can improve VLP interpretability for model behaviors. This strategy aims to develop VLPs that are both robust and interpretable, improving the reliability in security-sensitive applications

**Causal Inference.** Causal inference represents a promising research direction in multimodal intelligence. It provides interpretable explanations for predictions to increase trustworthiness and enable effective human-AI interaction. In multimodal intelligence, causal inference aims to explore relationships between cause and effect, how different modalities influence each other, and how integrated information from both modalities contributes to decision-making in VLPs. To assess causal reasoning fairly and accurately, CELLO (Chen et al., 2024a) introduces a benchmark dataset to comprehensively examine causality capabilities in LVLMs across four aspects: discovery, association, intervention, and counterfactual reasoning. Experimental investigations reveal notable limitations in the causal capabilities of LVLMs, while also demonstrating improved causal performance using the proposed chain-of-thought prompting strategy. Multimodal causality can contribute to robust learning in VLPs by distinguishing between genuine and adversarial examples through causal relationships.

## 7.2  Multimodal Attacks

**Adversarial Tactics.** VLPs employ attention mechanisms across modalities over the sequence dimension. Modal entities are mutually interconnected through dot-product similarity, enabling each element to attend to others across modalities. Building upon this theoretical foundation, VLP fusion strategies aim to provide multimodal contextualized representations, thereby enhancing cross-modal learning techniques (Liang et al., 2022). Multimodal attacks that simultaneously target images and text have been shown to be more effective than on a single modality attacks. These adversarial tactics exploit the alignment and fusion to disrupt the semantic correspondence between modalities. Understanding and mitigating these multimodal attacks by crafting cross-modal perturbations can substantially contribute to the development of more security multimodal systems.

**Adversarial Transferability.** Adversarial transferability refers to the phenomenon where adversarial examples generated from a source model can effectively misclassify different unseen target models. For VLPs, this problem entails investigating transferable adversarial attacks across the VLP family and identifying factors that contribute to attack transferability, which has received limited attention in existing research. Luo et al. (2024a) address this issue by simultaneously optimizing image inputs and multiple prompts in opposite directions while updating image perturbations to achieve cross-prompt adversarial transferability. Gao et al. (2025) further propose a diversification strategy for the intersection regions along the adversarial trajectory, which aims to increase transferability across various VLPs. These suggested directions can help improve adversarial robustness against evolving adversarial tactics and develop more generalizable defense strategies.

**Benchmarks.** Numerous safety-related benchmarks have been proposed to evaluate adversarial robustness in LVLMs. HADES (Li et al., 2024c) contributes a dataset comprising 750 harmful text-image pairs distributed across five distinct scenarios, using ChatGPT for instruction modification and diffusion models for harmful image generation. SafeBench (Gong et al., 2023b) creates a safety benchmark covering 10 safety topics with 500 harmful queries and images. MM-SafetyBench (Liu et al., 2024g) investigates how LVLMs perform under query-relevant images with malicious key phrases transformed into images comprising typography approaches. This benchmark proposes effective safety prompt strategies to enhance LVLM resilience and includes 13 scenarios containing 5,040 text-image pairs for image-based jailbreak evaluation, assessing 12 state-of-the-art LVLMs by those equipped with safety alignment mechanisms. JailBreakV-28K (Luo et al., 2024b) collects 2,000 harmful queries with 16 safety policy adversarial scenarios and contains 28,000 jailbreak text-image pairs. This dataset is strategically divided into two categories: 20,000 text-based transferability cases of LLM jailbreak techniques to LVLMs and 8,000 image-based LVLM jailbreak attacks, highlighting critical vulnerabilities. Cheng et al. (2024) introduce a typographic benchmark dataset for evaluating the vulnerability of LVLMs to existing typographic attacks, where malicious text embedded in images disrupts vision encoders, emphasizing the need for improving resistance to typographic vulnerabilities.

### 7.3 Adversarial Defense

**Adversarial Training.** Adversarial training (Madry et al., 2018; Bai et al., 2021) has received considerable attention and is generally considered one of the most effective adversarial defense strategies, although its effectiveness is limited by the characteristics of specific datasets and tasks. While adversarial training enhances adversarial robustness by augmenting standard training procedures, it negatively impacts generalization performance. Compared to standard training, adversarial training is computationally expensive due to the inner maximization requirement for adversarial loss through applying perturbations in the fusion embedding space. Gan et al. (2020) propose adversarial training to improve performance without incurring additional computational usage by creating adversarial perturbations in the multimodal embedding space. However, this approach focuses on improving model performance rather than defending against adversarial attacks. One natural extension to adversarial training is to create general-purpose VLP systems that do not rely heavily on task-specific properties, thereby reducing the negative impact on generalization performance without incurring significant theoretical limitations or computational costs. One direction is modality-decoupled adversarial training. Instead of adversarially retraining the entire LVLM, a robust vision encoder can be integrated while keeping the LLM backbone frozen. A question is whether replacing the original vision encoder with FARE-CLIP (Schlarmann et al., 2024) can reduce the attack success rate of visual jailbreaks (Qi et al., 2024) without retraining the LLM backbone. Another question is whether this protection transfers across LVLM architectures that use compatible vision encoders.

**Certified Robustness.** In preliminary explorations, we emphasize that adversarial robustness serves as a meaningful additional comparison point beyond standard evaluations (Carlini et al., 2019). Relying solely on empirical adversarial robustness evaluations is insufficient for reliable deployment in real-world scenarios. One of the key challenges in this area is the lack of certified robustness guarantees. Certified robustness aims to formally ensure that model outputs remain consistent across both clean and perturbed inputs, providing theoretical guarantees of robustness. The theoretical understanding of robust DNNs remains inadequate, and this limitation extends to even more complex VLPs. Randomized smoothing (Cohen et al., 2019) provides the most scalable approach to certification for large models, constructing a smoothed classifier

that is provably robust within an $\ell_2$-ball by predicting based on the majority vote under Gaussian noise. For instance, Nirala et al. (2024) introduce open vocabulary verification to certify the robustness of CLIP against adversarial attacks using incremental randomized smoothing. This approach leverages novel prompts as perturbations of nearby classifiers in the certification process using caching-based acceleration. In certified robustness research, the primary goal is to develop certification and verification approaches serving as quality assurance in response to perturbed data. Therefore, it is essential to verify VLPs to ensure they deliver acceptable performance under uncertain adversarial conditions. An important open problem is extending certification beyond $\ell_2$-ball guarantees to the $\ell_\infty$ threat model and, more importantly, to the *joint* vision-language input space. For example, certifying that a VLP-based VQA model preserves its answer under any combination of bounded, visually imperceptible image perturbations and meaning-preserving paraphrases of the question requires certification across both continuous visual and discrete textual spaces. For LVLMs, similar guarantees should focus on preserving task-relevant or safety-critical behavior rather than requiring identical generated responses. General multimodal certification frameworks capable of providing such joint guarantees remain underdeveloped.

**Adversarial Purification.** The diffusion-based generative models has introduced purification methods that are model-agnostic and without the need for task-specific retraining and architecture modifications (Shi et al., 2020; Yoon et al., 2021). Adversarial purification (Wu et al., 2022) is a safeguard that removes adversarial perturbations by generating purified inputs comparable to the original inputs, ensuring correct predictions by victim models. One key advantage of this approach is that it reduces the requirement for deeper understanding of model internals and defends against unseen threats without identifying specific adversarial attack types. DiffPure (Nie et al., 2022) establishes diffusion models as an effective purification backbone for unimodal vision tasks. A natural extension is semantics-preserving multimodal purification: jointly denoising the visual input while conditioning on the paired textual query, so that purification does not strip away visual details that are relevant to the downstream task. Recent work on VLP adversarial purification (Qi et al., 2024) demonstrates implementation in a plug-and-play manner to counter visual adversarial examples. This process purifies visual adversarial examples by reconstructing inputs through generative models that guide them back onto the data manifold, offering a flexible and modular approach that can be integrated into existing VLPs before prediction.

**Prompt Tuning**. Jailbreak prompts (Deng et al., 2023) are specially designed to bypass service provider constraints by exploiting AI alignment safeguards or other confinement measures. Additionally, prompt injection (Zhang et al., 2024b; Kan et al., 2023) focuses on controlling inputs by inserting adversarially constructed prompts. Conversely, prompt techniques can also be adapted for adversarial defense, by crafting robust prompts that resist jailbreak attempts and mitigate prompt injection. Inspired by advancements in prompt engineering, prompt tuning defense is a strategy designed to improve the adversarial robustness of VLPs without retraining models. By optimizing input prompts without altering model architectures or parameters, this methodology effectively mitigates adversarial vulnerabilities and provides a lightweight defense mechanism for existing VLPs. Chen et al. (2023a) propose a class-wise adversarial visual prompting (C-AVP) approach that generates class-specific visual prompts to enhance adversarial robustness. C-AVP benefits from ensemble prompts and optimized interrelationships. This approach overcomes the limitations of universal prompts for robust learning against adversarial perturbations. Devoting research effort to exploring defense strategies such as prompt tuning and adversarial prompt detection is an important future research direction. Whether defensive prompts trained on one attack family generalize to unseen adversarial and jailbreak attacks without raising refusal rates on benign queries remains an open and largely untested question.

### 7.4 Safety Enhancement

**Machine Unlearning.** Machine unlearning is an emerging paradigm that removes particular training data from machine learning models without requiring complete retraining. In the context of LVLMs, this capability allows for the efficient elimination of adversarial data while preserving overall model performance (Chakraborty et al., 2024). Therefore, machine unlearning techniques offer a promising approach for mitigating adversarial and jailbreak attacks. Specifically, machine unlearning provides a safeguard mechanism to revoke certain learned capabilities by removing adversarial or sensitive data from the training

process. As multimodal machine unlearning remains in its early stages, this field involves understanding how removing sensitive or private information impacts fused multimodal representations while ensuring consistent model performance after unlearning. Whether multimodal unlearning can remove the association between a visual trigger and a malicious response without erasing benign concepts shared across visual and textual modalities remains an open question.

**Model Editing.** Although the pretraining approach has proven to be advantageous for multimodal tasks, a limitation of LVLM pretraining lies in its dependence on large-scale datasets and substantial computational power. This raises the challenge of maintaining accurate and current knowledge without incurring substantial retraining costs. Model editing refers to manipulating the behavior of models in specific domains by efficiently modifying model parameters or integrating auxiliary modules while preserving the original and unaltered models. Recent research in model editing aims to address undesirable outputs while preserving the integrity of interpretations and without degrading performance on unrelated inputs. For example, Geva et al. (2022) introduce model editing to suppress potentially harmful text generation. And MMEdit (Cheng et al., 2023) presents a multi-modality model editing benchmark that achieves high fidelity across various vision-language tasks under three evaluation principles: reliability, locality, and generality. Thus, model editing can potentially mitigate toxic and illegal knowledge stored in LLMs in response to certain prompts, which is vital for enhancing the security and robustness of LVLMs. Whether model editing can suppress a specific harmful multimodal behavior without degrading locality, generality, or robustness on unrelated image-text inputs has not been established.

**Mitigating Hallucination.** There is growing concern regarding hallucination defined as the generation of content that is unfaithful to the input. Specifically, LLMs tend to hallucinate undesired text (Ji et al., 2023), and LVLMs generate nonexistent objects in images (Li et al., 2023c). Observe–Reason–Critique–Act (ORCA) (Yu et al., 2025) introduces a training-free agentic reasoning framework that addresses both hallucination mitigation and adversarial robustness in LVLM test time. It leverages lightweight external vision models and performs cross-model consistency verification to identify and correct unreliable predictions. This understanding helps develop more sophisticated hallucination detection and defense mechanisms against adversarial attacks. Whether cross-modal consistency checking can reliably distinguish naturally occurring hallucinations from adversarially induced hallucinations without access to model gradients, which remains unclear.

# 8 Conclusion

This review provides a systematic overview of adversarial attacks and defenses for VLPs based on three architectural families: VLFP, VLCL, and LVLM. It covers adversarial perturbations, prompt injection, jailbreaks, and backdoor attacks. It also reviews adversarial training, prompt-based defenses, input purification, backdoor mitigation, safety alignment, and certified robustness. We further identify open challenges and future research directions in VLP adversarial robustness. Ultimately, we hope this review serves as a useful reference for understanding the adversarial vulnerabilities of VLPs and as a starting point for future work in this rapidly evolving area.

## Broader Impact Statement

This review consolidates published research on adversarial attacks and defenses for vision-language pretraining models. We make three clarifications regarding its potential societal impact.

**No novel attacks are introduced.** All attack methods reviewed in this survey are drawn from peer-reviewed or publicly available preprints; this work introduces no new attack techniques. Our intent is to synthesize and analyze the existing literature to inform both researchers and practitioners.

**Dual-use acknowledgment.** Some of the attack methods catalogued here—including jailbreak methods (GCG, UMK, FigStep), physical backdoor attacks (BadVLMDriver) and data poisoning methods (Shadowcast)—carry dual-use potential. Presenting these techniques in a unified survey makes them more accessible. We note that consolidating the defense literature alongside the attack literature is itself a safety

contribution: it equips security practitioners with the knowledge needed to assess and mitigate these threats in deployed systems.

**Responsible disclosure norms.** The adversarial machine learning community has adopted norms of responsible disclosure. All works surveyed here were published in accordance with these norms. We encourage researchers building on this survey to follow coordinated vulnerability disclosure practices when evaluating adversarial threats in real-world systems and to prioritize publishing defense findings alongside attack findings.

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
