# OpenReview forum: "Adversarial Attacks and Defenses in Vision-Language Pre- training: Techniques, Challenges and Opportunities"
_TMLR — Decision pending for TMLR_

### Review · Reviewer_AqDK · 2026-05-03

**Summary Of Contributions:**

This manuscript surveys adversarial attacks and defenses for vision-language pretraining (VLP) models, organized around an architecture-centric taxonomy that partitions the literature into three families:
* Vision-Language Fusion Pretraining (VLFP),
* Vision-Language Contrastive Learning (VLCL),
* Large Vision-Language Models (LVLMs).

The authors review attack categories (prompt injection, adversarial perturbations, jailbreak, and backdoor attacks) and pair them with corresponding defense families (adversarial contrastive fine-tuning, adversarial prompt tuning, backdoor defense, jailbreak defense, certified robustness, and safety alignment). Sections 4 and 5 enumerate representative methods in tabular form (Tables 4 and 5), Section 6 connects the discussion to downstream applications (VQA, captioning, retrieval, reasoning, navigation), and Section 7 outlines future directions across model architecture, interpretability, causal inference, and other aspects.

Key Strength:
1. The architectural taxonomy (VLFP / VLCL / LVLM) is a reasonable organizing principle that distinguishes this work from prior LLM-centric surveys [1], [2].
2. The coverage of LVLM-specific threats, particularly cross-modal jailbreak and backdoor attacks, is current to academics.
3. Tables 3–5 are useful at-a-glance references, and the formal threat-model decomposition in Sections 3.1–3.3 (Eqs. 1, 7–8) provides a unifying notation that prior surveys generally omit.

Key weaknesses:
1. The manuscript shows signs of incomplete editorial preparation, including a broken cross-reference ("Section ??" in §3.3), and duplicated sentence pairs ("The adversarial robustness of VLPs in real-world...") in §7.
2. The proposed taxonomy is presented as the central contribution but is only weakly differentiated from the architectural decompositions already used in [3] and [4].
3. The survey is descriptive but rarely comparative. Reader gains a list of methods but limited insight into their relative effectiveness. For example, Tables 4 and 5 catalogue dozens of attacks and defenses with columns for "Strategy" and "Key Contribution," but neither the tables nor the surrounding paragraphs report the quantitative outcomes that would let a reader rank method or understand the trade-offs between them.
4. Figures 1 and 3 are conceptual block diagrams. The paper lacks any concrete visual walkthroughs of how each attack type modifies inputs or alters model behavior, despite this being the most common reader expectation in adversarial ML surveys ([5], [6]).
5. The defense literature for VLFPs is essentially declared absent in §5 without further critical discussion of why this gap exists.

[1] Shayegani, Erfan, Yue Dong, and Nael Abu-Ghazaleh. "Jailbreak in pieces: Compositional adversarial attacks on multi-modal language models." arXiv preprint arXiv:2307.14539 (2023).
[2] Jin, Haolin, et al. "From llms to llm-based agents for software engineering: A survey of current, challenges and future." arXiv preprint arXiv:2408.02479 (2024).
[3] Liu, Xiaogeng, et al. "AutoDAN: Generating Stealthy Jailbreak Prompts on Aligned Large Language Models." Proceedings of the Twelfth International Conference on Learning Representations, ICLR, 2024. OpenReview, openreview.net/forum?id=7Jwpw4qKkb.
[4] Zhang, Tingwei, et al. "Adversarial Illusions in Multi-Modal Embeddings." Proceedings of the 33rd USENIX Security Symposium, USENIX Association, 2024. arXiv, arxiv.org/abs/2308.11804.
[5] Akhtar, Naveed, and Ajmal Mian. "Threat of Adversarial Attacks on Deep Learning in Computer Vision: A Survey." IEEE Access, vol. 6, 2018, pp. 14410–14430. IEEE Xplore, https://doi.org/10.1109/ACCESS.2018.2807385.
[6] Chakraborty, Anirban, et al. "A Survey on Adversarial Attacks and Defences." CAAI Transactions on Intelligence Technology, vol. 6, no. 1, Mar. 2021, pp. 25–45. Wiley Online Library, https://doi.org/10.1049/cit2.12028.

**Additional Comments:**

I want to acknowledge that the authors have taken on this timely topic. Consolidating the adversarial robustness literature across three distinct VLP families is no small task, and the effort to unify notation across attack settings (Eqs. 1–8) is genuinely appreciated. The theoretical derivations thoughout these equations are also unified and solid. This level of formalization is often missing in survey papers and adds real pedagogical value.

A few additional observations that may help strengthen the next revision:
1. The architecture-based taxonomy would land more strongly if the paper demonstrated at least one concrete case where the same attack behaves differently across the three families
2. The authors may also consider reordering Sections 6.1 and 6.2 so that downstream tasks are introduced before the attack/defense sections and expanding the brief conclusion (§8) with one or two distilled headline findings to give readers a crisp takeaway.

Overall, I see a solid foundation here, and with a careful revision addressing the points raised above, I believe this work could make a meaningful contribution to the field.

**Audience:**

Yes

**Audience Explanation:**

Adversarial robustness in multimodal foundation models is an active and growing concern within the TMLR community. The topic intersects machine learning, computer vision, NLP, and AI safety.

The architectural taxonomy is a useful pedagogical lens, particularly for researchers entering the area who need to understand why attacks designed for CLIP-style contrastive encoders do not always transfer to fusion-based or instruction-tuned multimodal LLMs. The consolidated reference tables (Tables 3–5) and the unified notation in Section 3 have practical value as a starting point for newcomers in the field and as a reference index for established researchers.

Despite the issues identified above, the work fills a niche that is currently scattered across narrower surveys (LLM-only, jailbreak-only, or trustworthiness, etc.), and a properly revised version would be cited by the multimodal-safety community.

**Broader Impact Concerns:**

The manuscript currently does not include a Broader Impact Statement.

Given that the survey describes operational jailbreak and backdoor methodologies, including dual-use techniques such as Universal Master Key (UMK), and BadVLMDriver, etc., some targeting autonomous-driving systems, a Broader Impact Statement is preferred. I recommend the authors add a short section that:
a. acknowledges the dual-use nature of the catalogued attacks,
b. clarifies that all reviewed methods are drawn from already published peer-reviewed work and that no novel attack is introduced here,
c. discusses the norms of responsibility disclosure typical in adversarial ML and the safety value of consolidating defense literature with attack literature.

**Claims And Evidence:**

No

**Claims Explanation:**

In this survey paper, the central claims are:
1. that the proposed taxonomy is novel and useful,
2. that the coverage is comprehensive,
3. that the analysis identifies trends and gaps.
The current manuscript supports these claims only partially.

A more detailed review of these claims can be found below:
1. Novelty of taxonomy is overstated. In Table 1, authors assert that this survey provides comprehensive attack and defense coverage with explicit VLP/LVLM specificity, in contrast to prior work. However, [1] already partitions attacks by model family while mentioning the 2 major defense types, and [2] covers defenses across LVLM trustworthiness dimensions including safety and robustness. The differentiation in §1.3 would be more credible with a side-by-side mapping of which specific methods are covered uniquely here in this paper versus elsewhere, instead of the broader comprehensive/partial matrix form.
2. Comprehensiveness is uneven. Coverage of LVLM jailbreak attacks is reasonable, but several influential lines of work are still absent or underrepresented. For example, on the attack side, Greedy Coordinate Gradient (GCG, [3]), AutoDAN [4], and more query-related attacks beyond the cited entry, etc.; and on the defense side, RobustCLIP variants beyond FARE-CLIP, DiffPure adaptations to multimodal settings [5], and certified smoothing for multimodal classifiers [6] deserve at least brief mentioning. Without an explicit search protocol, it is difficult for a reader to assess what was systematically excluded.
3. Analytical depth is limited. Tables 4 and 5 catalog "Strategy" and "Key Contribution" but report no quantitative outcomes. There appear no attack success rates, no clean/robust accuracy trade-offs, no compute budgets, and no benchmark identifiers. As a result, the paper cannot substantiate its Section 4.4 or Section 5.3 claim that the field has produced an actionable understanding of which methods work, when, and why.
4. There are a few editorial errors, detailed in "Requested Changes" section, that weakens the authors' claims.


[1] Daizong Liu, Mingyu Yang, Xiaoye Qu, Pan Zhou, Yu Cheng, and Wei Hu. A survey of attacks on large
vision-language models: Resources, advances, and future trends. arXiv preprint arXiv:2407.07403, 2024b.
[2] Haotian Liu, Chunyuan Li, Yuheng Li, and Yong Jae Lee. Improved baselines with visual instruction tuning.
In Proceedings of the IEEE/CVF Conference on Computer Vision and Pattern Recognition, pp. 26296
26306, 2024d.
[3] Zou, Andy, et al. "Universal and transferable adversarial attacks on aligned language models." arXiv preprint arXiv:2307.15043 (2023).
[4] Liu, Xiaogeng, et al. "AutoDAN: Generating Stealthy Jailbreak Prompts on Aligned Large Language Models." Proceedings of the Twelfth International Conference on Learning Representations, ICLR, 2024. OpenReview, openreview.net/forum?id=7Jwpw4qKkb.
[5] Nie, Weili, et al. "Diffusion Models for Adversarial Purification." Proceedings of the 39th International Conference on Machine Learning, vol. 162, PMLR, 2022, pp. 16805–16827. proceedings.mlr.press/v162/nie22a.html.
[6] Yang, Greg, et al. "Randomized Smoothing of All Shapes and Sizes." Proceedings of the 37th International Conference on Machine Learning, vol. 119, PMLR, 2020, pp. 10693–10705. proceedings.mlr.press/v119/yang20c.html.

**Requested Changes:**

Politely requesting a few changes that would significantly improve the quality of this survey paper manuscript, listed below:

1. Editorial pass and figure remediation. Resolve the broken "Section ??" cross-reference in §3.3, eliminate duplicated paragraphs in §7 (the "The adversarial robustness of VLPs in real-world deployments" sentence appears twice, and perform a thorough proofread edit.
2. Concrete visual walkthroughs for each attack family. Figures 1 and 3 are block-level overviews. The paper would benefit substantially from per-family illustrative examples showing original input → perturbation → adversarial input → model output, for at least one representative method in one or multiple of prompt injection, adversarial perturbation, jailbreak, and backdoor attacks. Existing surveys in adjacent areas (e.g., [1]) succeed in part because of such concrete worked examples. This is essential for the paper to function as a teaching reference for others.
3. Component-level comparison and analysis tables. Tables 4 and 5 should be augmented with quantitative columns where the original papers report them. For example, metrics such as target benchmark, victim model, attack success rate or robust accuracy, and perturbation budget. Without these, the survey cannot support claims about which methods are state-of-the-art, and the reader cannot make informed methodological choices. A small ablation-style table examining how the same defense (e.g., adversarial contrastive tuning) performs across VLFP / VLCL / LVLM backbones drawn from already published numbers would meaningfully sharpen the survey thesis.
4. Sharper differentiation from prior surveys. Supplement the qualitative/Partial comparison in Table 1 with a method-level coverage matrix indicating which specific attacks and defenses are reviewed exclusively here versus elsewhere. This is the most direct way to substantiate the novelty claim and make the comparison clearer to the readers.
5. Documented search and inclusion protocol. Section 1.3 mentions a three-stage screening but provides no PRISMA-style flow diagram, no inclusion and exclusion criteria, and no count of papers screened versus included.

[1] Chakraborty, A., Alam, M., Dey, V., Chattopadhyay, A., & Mukhopadhyay, D. (2021). A survey on adversarial attacks and defences. CAAI Transactions on Intelligence Technology, 6(1), 25–45.

---

> ### Author Response · Authors · 2026-06-28
> **We thank the reviewer for the constructive feedback. We strengthened the taxonomy positioning, analytical comparisons, quantitative tables, visual attack walkthroughs, literature coverage, search protocol, VLFP defense analysis, manuscript organization, and broader impact discussion.**
>
> We sincerely thank the reviewer for the detailed and constructive feedback. We have substantially revised the manuscript as follows.
>
> ### 1. Editorial Corrections
> - We resolved the broken cross-reference in the previous Section 3.3 and removed the duplicated text in Section 7.
> - We conducted an additional proofreading pass.
>
> ### 2. Taxonomy Positioning
> We thank the reviewer for this helpful comment. In the revised manuscript, we have softened the wording from a “novel taxonomy” to an “architecture-aware taxonomy.” Specifically, we now explain that our taxonomy is organized around three representative VLP paradigms: VLFP, VLCL, and LVLMs.
>
> ### 3. Strengthened Analysis and Comparisons
> Thank you for encouraging deeper analysis. We strengthened the synthesis in Sections 3, 5, and 6.
>
> - **Section 3** compares adversarial impacts on discriminative and generative tasks and discusses cross-task transferability, high-stakes risks, and benchmark gaps.
> - **Section 5** compares architecture-specific attacks: VLFP attacks exploit cross-modal fusion, VLCL attacks disrupt shared embedding spaces, and LVLM attacks target generative and safety-alignment mechanisms. It also identifies the shift toward transferable and universal attacks.
> - **Section 6** compares defense strategies, robustness/accuracy trade-offs, and the shift from reactive defenses toward purification, safety alignment, and multimodal machine unlearning.
>
> ### 4. Quantitative Attack and Defense Comparisons
> Thank you for highlighting the need for clearer presentation in the tables. We have substantially revised Tables 4 and 5 to provide more informative comparisons of the reviewed methods.
>
> - **Table 4 (attacks)** now reports the Attack Method, Attacked Modality, Strategy, Benchmark, Victim Model, Key Result, and Key Contribution. The methods are grouped into VLFP, VLCL, and LVLM categories. The Key Result column presents representative quantitative outcomes where reported in the original studies.
>
> - **Table 5 (defenses)** adopts a similar structure and reports the Defense Method, Defended Modality, Strategy, Benchmark, Victim/Base Model, Key Result, and Key Contribution. The methods are grouped into VLCL and LVLM categories. The absence of a VLFP group reflects the scarcity of VLFP-specific defenses, as discussed in Section 6.3. The Key Result column similarly summarizes representative quantitative outcomes reported in the original studies.
>
> ### 5. Concrete Visual Attack Processes
> Thank you for suggesting concrete visual wokflow. We added four figures showing how attacks modify inputs and model behavior:
>
> - **Figure 4:** visual and textual perturbations leading to incorrect predictions;
> - **Figure 5:** trigger implantation and activation in backdoor attacks;
> - **Figure 6:** textual prompt-injection jailbreaks that bypass refusal mechanisms; and
> - **Figure 7:** multimodal jailbreaks delivered through textual and visual channels.
>
> ### 6. VLFP Defense Gap
> Thank you for highlighting this research gap. We expanded Section 6.3 to explain that VLFP-specific defenses are limited. We also discuss how the shift toward VLCL models and LVLMs has reduced attention to VLFP-specific defenses.
>
> ### 7. Expanded Coverage of Attack and Defense Methods
> Thank you for this observation.
> - In Section 5.3, we expanded the discussion of **GCG** and **AutoDAN** as representative text-space jailbreak attacks, together with additional transfer- and query-based attacks against LVLMs.
>
> - In Section 6.1, we expanded the coverage of robust CLIP-based defenses, including **Sim-CLIP, MMCoA, PMG-AFT, TeCoA,** and **LAAT**, as well as certified robustness methods based on randomized smoothing, including **PromptSmooth** and **Open Vocabulary Certification**. Section 6.2 also discusses **FARE-CLIP** and **DiffPure-based multimodal input purification**.
>
> ### 8. Literature Search and Inclusion Criteria
> Thank you for highlighting the literature-selection process. Our review was designed as a broad, cross-domain literature survey rather than a review following the PRISMA reporting framework. Therefore, stage-level counts were not recorded in the format required for a PRISMA flow diagram, and reconstructing them retrospectively could introduce inaccurate information. To address the reviewer’s underlying concern, we expanded Section 1.3 to report the publication period, target venues, databases, search terms, screening procedure, inclusion criteria, and the number of studies retained for the core synthesis.
>
> ### 9. Reorganization of Downstream Tasks and Applications
> Thank you for this suggestion. We reorganized the manuscript so that downstream tasks and the corresponding application impacts of adversarial attacks are now introduced in Section 3, before the attack and defense sections.
>
> ### 10. Broader Impact
> Thank you for highlighting this issue. We added a dedicated **Broader Impact Statement** to the revised manuscript.

---

### Review · Reviewer_zX59 · 2026-05-27

**Summary Of Contributions:**

This paper presents a survey of adversarial attacks and defenses for Vision-Language Pretraining (VLP) models. The authors propose an architecture-based taxonomy that categorizes VLP models into three families—VLFP, VLCL, and LVLM—and organize the existing attack and defense literature accordingly.

The main contributions claimed by the authors are as follows:
- An architecture-based taxonomy of recent VLP models, along with a structured overview of corresponding adversarial attacks and defense methods.
- A discussion of key challenges and open problems in adversarial robustness for VLP models, as well as potential future research directions.

**Audience:**

No

**Audience Explanation:**

Adversarial robustness in vision-language and multimodal models is a rapidly growing research area, and a survey that organizes the literature according to VLP architectures could certainly be useful for researchers entering the field, as well as for practitioners deploying these models in safety-sensitive applications.

However, given that recent AI systems (e.g., Claude- or ChatGPT-like models) have become increasingly effective at searching, summarizing, and organizing papers, I believe the bar for survey papers should now be higher. A survey paper should provide value beyond a straightforward enumeration of prior work and instead offer clear direction through deeper synthesis and analysis. In particular, as emphasized in the TMLR FAQ, a strong survey paper should provide genuinely new connections, in-depth analyses that reveal meaningful trends, and non-trivial open problems or future directions derived from such analyses.

In the current version, I do not believe the paper reaches that level of synthesis or insight. While it presents a broad categorization and summary of prior work, much of the discussion remains descriptive rather than analytical. As a result, I ultimately lean toward rejection.

**Broader Impact Concerns:**

Wrote above

**Claims And Evidence:**

No

**Claims Explanation:**

The TMLR FAQ states that survey papers should “draw new, previously unreported connections between several pieces of work in an area, and/or clearly highlight trends in the area, and/or suggest currently open problems.”
Therefore, I evaluated the paper along each of these axes:



(a) New connections: Limited.
In my understanding, the core contribution claimed by the authors is the analysis of adversarial attacks and defenses through the lens of architectural differences among VLP models. However, the proposed categorization itself is already well-established in the VLP literature. Therefore, the key question is whether the paper identifies architecture-specific challenges, shared assumptions, empirical patterns, or distinctive design trade-offs related to adversarial robustness. In the current version, I could not find a sufficiently deep analysis of these aspects. Instead, many sections read more like a straightforward enumeration of prior works rather than a synthesis that reveals new insights or conceptual connections.


(b) Trends: Discussed only at a surface level.
Although the paper summarizes existing methods and organizes them according to architectural categories and attack/defense types, the discussion does not sufficiently reveal broader research trends or meaningful evolution of the field. For example, the paper does not sufficiently provide analyses of how attack or defense strategies have shifted over time, which architectural paradigms are becoming dominant, or how robustness performance has quantitatively evolved across different generations of models. As a result, it is difficult for readers to extract a clear understanding of the major trends or trajectories in this research area.

(c) Open problems and future directions: Too generic.
Section 7 presents several future directions, including “Model Architecture,” “Adversarial Training,” “Model Editing,” “Mitigating Hallucination,” and “Certified Robustness.” However, the discussion remains at a high level of abstraction and does not lead to concrete or actionable research questions. In many cases, the section mainly reiterates that these areas remain underexplored, without proposing specific hypotheses, mechanisms, evaluation settings, or technical challenges that would meaningfully guide future work. Moreover, many of these directions are already widely recognized in the existing literature. Overall, I believe the main issue is that the preceding analysis is itself too shallow to naturally support deeper or more insightful future research directions.

**Requested Changes:**

1. The most important point, in my opinion, is that the paper should go beyond a survey that merely enumerates prior work along the VLP architectural axis. Instead, it should provide a more in-depth analytical perspective. To satisfy the three criteria discussed above (new connections, meaningful trends, and non-trivial open problems), stronger synthesis and deeper analysis are necessary throughout the paper. In particular, if architectural differences are presented as the main organizing principle, the paper should more clearly articulate what fundamentally changes of attack or defense across architectures. Moreover, at present, the categorization itself largely follows already well-established distinctions in the VLP literature. Therefore, the paper would benefit from analyses that go beyond architectural categorization alone and introduce additional axes of comparison or interpretation.

2. There are also several more minor issues:
- Table 3 formatting and categorization issues. The column structure in Table 3 appears inconsistent. Even assuming the “text encoder” and “vision encoder” columns are unintentionally shifted, it is still confusing that components such as “Q-former,” “Linear Projection,” and “Multi-modal Learning” under LVLM are categorized as pre-training tasks. These entries seem conceptually different from the other items in the table, and the taxonomy should be clarified.
- Insufficient discussion of the transition from unimodal to multimodal adversarial robustness. Since the paper focuses on multimodal attacks and defenses, it is natural that multimodal settings receive the main emphasis. However, I believe the survey should also more thoroughly discuss the foundations established in unimodal adversarial robustness literature and explicitly analyze what new challenges emerge in multimodal settings compared to unimodal ones. Currently, in my opinion, this transition and its implications are not sufficiently analyzed.

---

> ### Author Response · Authors · 2026-06-28
> **We softened the taxonomy claim, expanded Sections 4–7, added architecture-specific attack and defense comparisons, corrected Table 3, and clarified the new challenges arising in multimodal robustness.**
>
> We sincerely thank the reviewer for the critical and thoughtful feedback. We agree that the previous manuscript required deeper synthesis, clearer trends, and more actionable research directions. We have revised the manuscript as follows.
>
> ### 1. Reframing the Taxonomy Contribution
> Thank you for this important observation. We softened the wording from a **“novel taxonomy”** to an **“architecture-aware taxonomy.”** We acknowledge that VLFP, VLCL, and LVLM are established architectural categories. Our contribution is to use these categories as a common framework for analyzing how attack and defense assumptions, evaluation settings, and robustness limitations vary across model families. We clarified this positioning in Sections 1.2–1.3 and Table 1.
>
> ### 2. Architecture-Specific Connections and Comparisons
> Thank you for your valuable and constructive feedback.. We revised Section 5.4 to compare how attack mechanisms differ across VLP architectures.
> - **VLFP attacks** primarily exploit cross-modal fusion and attention, allowing perturbations in one modality to influence the fused representation.
> - **VLCL attacks** target the geometry of the shared image-text embedding space and can corrupt image-text associations through poisoning.
> - **LVLM attacks** additionally target modality connectors, autoregressive generation, and safety-alignment mechanisms.
>
> For defenses, Section 6.3 highlights the following architecture-specific differences:
> - **VLCL defenses** involve trade-offs among adversarial fine-tuning, prompt tuning, and certified robustness in terms of clean accuracy, robust accuracy, and computational cost.
> - **LVLM defenses** increasingly rely on retraining-free methods, although their robustness against adaptive white-box attacks remains insufficiently evaluated.
>
> ### 3. Research Trends and Empirical Evidence
> Thank you for highlighting the need to clarify the evolution of the field. We added explicit trend analysis to the revised summary sections.
>
> - Section 5.4 identifies the shift from task and model specific attacks toward transferable, universal and physical attacks.
> - Section 6.3 discusses the transition from reactive defenses against known attacks toward purification, safety alignment, and multimodal machine unlearning. It also explains the broader architectural shift in research attention from VLFPs toward VLCL models and LVLMs.
>
> Tables 5 and 6 now report benchmarks, victim/base models, and representative quantitative results from the original studies as the original studies use different tasks, threat models, perturbation budgets, and metrics, we report each result under its original experimental setting.
>
> ### 4.  Open Problems
> Thank you for pointing out that the previous future directions were too generic. We revised Section 7 to formulate more concrete technical questions and evaluation settings, including:
> - whether replacing an LVLM vision encoder with a robust encoder can reduce visual-jailbreak success without retraining the LLM backbone;
> - whether such protection transfers across LVLM architectures;
> - how to certify robustness jointly over continuous visual perturbations and discrete, meaning-preserving textual changes;
> - whether defensive prompts generalize to unseen attacks without increasing refusal rates on benign inputs;
> - whether multimodal unlearning can remove trigger-response associations without erasing related benign concepts; and
> - whether cross-modal consistency can distinguish natural hallucinations from adversarially induced hallucinations.
>
>
> ### 5. Correction and Clarification of Table 3
> Thank you for identifying the formatting and categorization problems in Table 3.
>
> - We corrected the column alignment and reorganized the table into the following fields: **Model, Language Encoder, Vision Encoder, Fusion/Modality Bridge, Training Objective,** and **Training Data**.
>
> - For LVLMs, components such as **Q-Former** and **Linear Projection** are now correctly placed under **Fusion/Modality Bridge**, while objectives such as vision-language alignment, multimodal language modeling, and instruction tuning are listed under **Training Objective**.
>
> ### 6. From Unimodal to Multimodal Robustness
>
> Thank you for highlighting this missing transition. We added a dedicated discussion at the beginning of Section 4. It reviews foundational unimodal attacks, including norm-bounded visual perturbations and discrete textual manipulations, and then explains the challenges introduced by multimodal models.
>
> These challenges include joint optimization across continuous visual and discrete textual spaces, attacks that propagate across modalities through fusion or alignment, and component-level robustness that does not guarantee end-to-end robustness. For example, a robust vision encoder does not address language-side vulnerabilities such as prompt injection. Together, these points explain why multimodal robustness is a distinct problem rather than a straightforward extension of unimodal robustness.

---

### Review · Reviewer_Fodt · 2026-06-01

**Summary Of Contributions:**

This paper surveys adversarial attacks and defenses for vision-language pretraining and related multimodal models. The paper organizes the literature around three model families, namely vision-language fusion pretraining, vision-language contrastive learning, and large vision-language models, and then reviews attack settings, attack methods, defense methods, application impacts, and future research directions. The topic is timely and relevant, and the manuscript covers a broad set of references across multimodal robustness and safety.

The main strength of the paper is its breadth and its attempt to structure a rapidly growing area under a unified narrative. The topic is important, and a survey in this space could be valuable to the community.

The main weaknesses are that the paper does not yet provide enough survey-specific synthesis, critical comparison, or conceptual insight beyond collecting and grouping prior works. In particular, the claimed “novel taxonomy” is not convincingly novel, the summary subsections are often too weak to synthesize the reviewed literature, and the core sections read largely as paper-by-paper listings rather than a survey that draws new connections, highlights strong trends, or surfaces new problems in a substantive way.

**Audience:**

Yes

**Audience Explanation:**

The topic is important and relevant to TMLR readers. Many researchers in multimodal learning, robustness, and trustworthy ML would be interested in a survey on adversarial attacks and defenses for vision-language models.

**Claims And Evidence:**

Yes

**Claims Explanation:**

I do not think the paper currently provides enough survey-specific synthesis to support its main claims. While the topic is timely and the paper covers a broad range of prior work, the core sections, especially Sections 4, 5, and 6, read mostly as sequential summaries of papers rather than a survey that draws new connections, highlights clear trends, or extracts deeper lessons from the literature.

I am also not convinced by the claim of a “novel taxonomy.” The main split into VLFP, VLCL, and LVLM is a reasonable organizational structure, but it largely follows standard architectural distinctions rather than introducing a clearly new conceptual framework. Relatedly, the section summaries are often too brief and do not sufficiently synthesize the material into key takeaways, tradeoffs, or open gaps.

There are also some clarity issues in presentation. Figure 3 is too dense and the text is unreadably small, and Figure 1 includes two dashed arrows whose meaning is not clearly explained. Overall, the paper is relevant and reasonably comprehensive, but in its current form it feels more like a catalog of existing papers than a survey that delivers substantial new insight for TMLR’s audience.

**Requested Changes:**

1. Critical: Add stronger synthesis in Sections 4, 5, and 6, including clearer trends, comparisons, and takeaways across papers.
2. Critical: Justify the “novel taxonomy” claim more convincingly, or soften it.
3. Critical: Improve the summary subsections so they synthesize the literature rather than briefly recap it.
4. Critical: Clarify the distinction between adversarial attacks, jailbreak attacks, and backdoor attacks.
5. Strengthening: Improve Figure 3, which is currently too dense and hard to read.
6. Strengthening: Explain the two dashed arrows in Figure 1.

---

> ### Author Response · Authors · 2026-06-28
> **We strengthened the core synthesis and summary sections, softened the taxonomy claim, clarified the distinctions among attack categories, redesigned Figure 3, and explained the visual connections in Figure 1.**
>
> We thank the reviewer for their constructive feedback and thoughtful suggestions. We have revised the manuscript as follows.
>
> ### 1. Stronger Synthesis
> Thank you for highlighting the need for deeper synthesis. We strengthened the analysis in Sections 4, 5, and 6.
> - **Section 4** now explains the transition from unimodal to multimodal adversarial robustness and analyzes how multimodal interactions introduce joint optimization, cross-modal alignment, and component-level vulnerabilities.
> - **Section 5.4** compares attacks across architectures and identifies the shift from task-specific attacks toward transferable and universal attacks. It also discusses gaps in cross-family transferability, physical-world robustness, and attacks on modality projection layers.
> - **Section 6.3** analyzes the VLFP defense gap, architecture-specific defense trade-offs, limitations of certified robustness, and the shift from reactive defenses toward purification, safety alignment, and multimodal machine unlearning.
>
> ### 2. Taxonomy Positioning
> Thank you for this important observation. We softened the wording from a **"novel taxonomy"** to an **"architecture-aware taxonomy."** We acknowledge that VLFP, VLCL, and LVLM are established architectural categories. Our contribution is not the architectural division itself, but its use as a common framework for comparing attack and defense strategies. We clarified this positioning in Sections 1.2–1.3 and Table 1.
>
> ### 3. Improved Summary Subsections
> Thank you for pointing out the weakness of the previous summaries. We rewrote the summary subsections around explicit analytical themes.
> - **Section 3.3** compares discriminative and generative tasks, cross-task transferability, high-stakes risks and benchmark gaps.
> - **Section 5.4** synthesizes architecture-specific attacks, transferability trends and remaining attack gaps.
> - **Section 6.3** compares defense assumptions, robustness/accuracy trade-offs, certification limitations and emerging proactive defenses.
>
> ### 4. Clarification of Attack Categories
> Thank you for identifying this ambiguity. We revised Section 5 to distinguish the three attack categories according to their mechanisms and objectives. **Adversarial perturbation attacks** modify visual, textual, or multimodal inputs, typically at inference time, to alter predictions or representations. **Jailbreak attacks** aim specifically to bypass safety alignment and elicit prohibited outputs; prompt injection and adversarial perturbations may serve as mechanisms for achieving this objective. **Backdoor attacks** implant trigger-dependent malicious behavior, typically during training, while preserving normal behavior on benign inputs. These definitions clarify the boundaries and relationships among the three categories.
>
> ### 5. Improved Figure 3
> Thank you for highlighting the readability issue. We redesigned Figure 3 by separating VLFP, VLCL, and LVLM architectures into three vertically arranged panels, enlarging the labels, simplifying the internal components and expanding the caption. The revised figure more clearly presents cross-modal fusion, contrastive alignment, modality connectors, and language-model components.
>
> ### 6. Explanation of Figure 1
> Thank you for pointing out this ambiguity. We revised Figure 1 and its caption to clarify the meanings of the dashed connections. The blue dashed arrow represents the delivery of manipulated textual and visual inputs to the target VLP. The yellow dashed connections indicate that the corresponding encoder, projection, and language-model components belong to the target model paths.